

# Changes in the geometry and strength of the Atlantic Meridional Overturning Circulation during the last glacial (20-50 ka)

Pierre Burckel[1], Claire Waelbroeck[1], Yiming Luo[2], Didier Roche[1,3], Sylvain Pichat[4], Samuel L. Jaccard[5], Jeanne Gherardi[1], Aline Govin[1], Jörg Lippold[5], François Thil[1]

[1]Laboratoire des Sciences du Climat et de l'Environnement, LSCE/IPSL, CEA-CNRS-UVSQ, Université Paris-Saclay, F-91191 Gif-sur-Yvette, France

[2]Dalhousie University, Department of Oceanography, 1355 Oxford Street, PO BOX 15000, Halifax, NS B3H 4J1, Canada

[3]Department of Earth Sciences, Earth and Climate Cluster, Faculty of Earth and Life Sciences, Vrije Universiteit Amsterdam, Amsterdam, The Netherlands

[4]Laboratoire de Géologie de Lyon (LGL-TPE), Ecole Normale Supérieure de Lyon, 46 allée d'Italie, 69007 Lyon, France

[5]Institute of Geological Sciences and Oeschger Centre for Climate Change Research, University of Bern, Baltzerstr. 1+3, CH-3012, Bern, Switzerland

*Correspondence to*: Pierre Burckel (pierre.burckel@lsce.ipsl.fr)

**Abstract.** We reconstruct the geometry and strength of the Atlantic Meridional Overturning Circulation during Heinrich Stadial 2 and three Greenland interstadials of the 20-50 ka period based on the comparison of new and published sedimentary $^{231}Pa/^{230}Th$ data with simulated sedimentary $^{231}Pa/^{230}Th$. We show that the deep Atlantic circulation during these interstadials was very different from that of the Holocene. Northern-sourced waters likely circulated above 2500 m depth, with a flow rate lower than that of the present day North Atlantic Deep Water (NADW). Southern-sourced deep waters most probably flowed northwards below 4000 m depth into the North Atlantic basin, and then southwards as a return flow between 2500 and 4000 m depth. The flow rate of this southern-sourced deep water was likely larger than that of the modern Antarctic Bottom Water (AABW). At the onset of Heinrich Stadial 2, the structure of the AMOC significantly changed. The deep Atlantic was probably directly affected by a southern sourced water mass below 2500 m depth, while a slow southward flowing water mass originating from the North Atlantic likely influenced depths between 1500 and 2500 m down to the equator.

## 1 Introduction

Greenland ice core records show that the last glacial climate repeatedly shifted between cold (stadial) and warm (interstadial) conditions (Johnsen et al., 1992). Greenland Stadials (GS) and Greenland Interstadials (GI) are the Greenland expressions of the characteristic millennial-scale Dansgaard-Oeschger events that represent cold and warm phases of the North Atlantic region, respectively (Rasmussen et al., 2014). GS typically lasted for several centuries, and were followed by a rapid warming of up to 15°C achieved in at most a couple of centuries (Kindler et al., 2014). The subsequent GI then lasted for several centuries to millennia, with Greenland temperatures slowly decreasing and leading to the onset of a new GS. During some of the GS, icebergs were released from high latitude northern hemisphere ice sheets into the North Atlantic Ocean, and their melting led to the deposition of ice rafted detritus on the seafloor, as observed in marine sediment cores (Heinrich, 1988). We refer to these periods as Heinrich Stadials (HS).

Changes in Atlantic Ocean circulation have long been suggested to impact Greenland temperatures (Broecker et al., 1985) and could be at the origin of the glacial millennial-scale variability. Indeed, there is much evidence for decreased North Atlantic deep-water formation and increased influence of southern-sourced deep waters in the Atlantic during Heinrich Stadials (Elliot et al., 2002; McManus et al., 2004; Skinner et al., 2003; Vidal et al., 1997). Moreover, climate models are able to reproduce



the bipolar seesaw pattern characterizing millennial-scale glacial variability through variations of the strength of the Atlantic Meridional Overturning Circulation (AMOC) in response to freshwater forcings (Ganopolski and Rahmstorf, 2001). However, recent studies show that a shallow circulation cell could have been still active during HS (Bradtmiller et al., 2014; Gherardi et al., 2009; Lynch-Stieglitz et al., 2014; Roche et al., 2014; Wary et al., 2015), suggesting that other mechanisms could be required to explain Greenland temperature millennial-scale variability. A better understanding of the vertical layout and flow rate of the water masses constituting the AMOC during the last glacial is therefore needed to assess the relationship between AMOC and glacial millennial-scale variability.

Sedimentary $(^{231}Pa/^{230}Th)_{xs,0}$ (activity ratio of $^{231}Pa$ and $^{230}Th$ unsupported by lithogenic and authigenic uranium and corrected from decay to the time of sediment deposition, Pa/Th hereafter) records were first used to assess the AMOC intensity during the Last Glacial Maximum (LGM) (Yu et al., 1996). Since then, Pa/Th records have been used in the Atlantic to infer changes in the intensity of the deep ocean circulation during HS (Böhm et al., 2015; Burckel et al., 2015; Gherardi et al., 2005, 2009; McManus et al., 2004). The comparison of simulated sedimentary Pa/Th values with core top Pa/Th data has shown that sedimentary Pa/Th reflects circulation intensity in the modern Atlantic Ocean (Lippold et al., 2011).

However, interpretation of sedimentary Pa/Th from a single sediment core might be complicated by the non-linear response of Pa/Th to circulation intensity changes (Luo et al., 2010; Thomas et al., 2006). Reconstructing present and past strengths of the AMOC is therefore best achieved by combining Pa/Th records from different water depths and latitudes (Gherardi et al., 2009; Lippold et al., 2011, 2012).

In this study, we present new sedimentary Pa/Th data from a deep sediment core recovered from the Brazilian margin, and from an intermediate depth core from the mid-latitude North Atlantic. We then compare last glacial Pa/Th records from different water depths and latitudes with Pa/Th values simulated using a simple 2D box model (Luo et al., 2010) forced by various streamfunctions. The streamfunctions were simulated with the Earth System model iLOVECLIM under different climatic conditions (Roche et al., 2014). Results of this comparison allow us to constrain the geometry and strength of the AMOC during GI-3, -8, -10 (Rasmussen et al., 2014) and HS2. We chose to focus our study on HS2 and the interstadials surrounding HS2 and HS4 as these periods are associated with very different ice-sheet volumes (Lambeck and Chappell, 2001).

## 2 Material and methods

### 2.1 Sediment cores

Sediment cores MD09-3257 (04°14.69'S, 36°21.18'W, 2344 m water depth) and MD09-3256Q (03°32.81'S, 35°23.11'W, 3537 m water depth) were recovered from the Brazilian margin during R/V Marion Dufresne cruise MD173/RETRO3 (Fig. 1). Improved recovery of deep-sea sediments with no or little deformation of sediment layers was achieved during this coring cruise thanks to the systematic use of the CINEMA software (Bourillet et al., 2007; Woerther and Bourillet, 2005). This software computes the amplitude and duration of the elastic recoil of the aramid cable, and the piston displacement throughout the coring phase, accounting for the length of the cable (water depth) and total weight of the coring system. The length of the coring cable is indeed of primary importance regarding the deformation rate of the 'Calypso' long piston cores (Bourillet et al., 2007; Skinner and McCave, 2003).

Core GeoB3910 (04°14.7'S, 36°20.7'W, 2362 m water depth) (Jaeschke et al., 2007) was recovered from approximately the same position and depth as core MD09-3257, during Meteor cruise M34/4. Hereafter we refer to both GeoB3910 and MD09-3257 as intermediate equatorial cores and to MD09-3256Q as the deep equatorial core. At present, the Brazilian margin at these depths is bathed by the North Atlantic Deep Water (NADW) (Fig. 1). Because the Brazilian margin is affected by western



boundary currents (Rhein et al., 1995), these sediment cores are ideally located to observe changes in the strength and extent of the intermediate and deep AMOC water masses (Schott, 2003).

Sediment core SU90-03 (40°30.3'N, 32°3.198'W, 2475 m) was recovered from the northern margin of the subtropical gyre (Chapman et al., 2000). Its location in the mid-latitude North Atlantic provides information on changes in NADW production
rates that could not be deduced from the sole equatorial depth transect.

We compare these Pa/Th records with published records from other Atlantic cores that span the 20-50 ka period: ODP Leg 172 site 1063 (33°41'N, 57°37'W, 4584 m, ODP1063 hereafter) (Böhm et al., 2015), MD02-2594 (34°43'S, 17°20'E, 2440 m) (Negre et al., 2010) and V29-172 (33°42'N, 29°22.98'W, 3457 m) (Bradtmiller et al., 2014) (Fig. 1, Table S1).

### 2.1.1 Benthic δ13C

The stable carbon isotopic composition ($\delta^{13}$C) of the epifaunal benthic foraminifer *Cibicides wuellerstorfi* has been shown to record the $\delta^{13}$C of bottom-water dissolved inorganic carbon (DIC) with minor isotopic fractionation (Duplessy et al., 1984; Zahn et al., 1986). Initial DIC isotopic concentration is acquired by a water mass in its formation region by surface productivity (which consumes $^{12}$C therefore increasing dissolved $\delta^{13}$C) and temperature dependent air-sea interactions (Lynch-Stieglitz et al., 1995; Rohling and Cooke, 2003). DIC $\delta^{13}$C then evolves as deep water ages, because the constant export of $^{12}$C-enriched
biogenic material that is remineralized at depth leads to the decrease of the DIC $\delta^{13}$C along the flow path of the water mass. As DIC $\delta^{13}$C largely follows water mass structure and circulation in the modern ocean, *C. wuellerstorfi* $\delta^{13}$C has been used to trace water masses, with a decrease in *C. wuellerstorfi* $\delta^{13}$C being interpreted as a decrease in bottom water ventilation, and conversely (e.g. Duplessy et al., 1988). However, the information on bottom water ventilation embedded in *C. wuellerstorfi* $\delta^{13}$C is complicated by the impact of changes in surface water $\delta^{13}$C, marine biological productivity and continental biomass
changes.

Because LGM $\delta^{13}$C values are higher in northern sourced waters (1.5 ‰) than in southern sourced waters (<-0.2 ‰)(Curry and Oppo, 2005), we interpret a decrease in *C. wuellerstorfi* $\delta^{13}$C values at the equatorial sites as an increase in the time elapsed since the water mass was last in contact with the atmosphere or as an increased influence of nutrient-rich southern sourced deep waters.
Core MD09-3256Q benthic foraminifer *C. wuellerstorfi* were handpicked in the size fraction higher than 250μm, washed with methanol in an ultrasonic bath, and then roasted in glass vials at 380°C under vacuum for 45 min. *C. wuellerstorfi* $\delta^{13}$C (expressed in ‰ VPDB) was measured at LSCE (Gif-sur-Yvette) using an Elementar Isoprime mass spectrometer. VPDB is defined with respect to NBS-19 calcite standard ($\delta^{18}$O = -2.20 ‰ and $\delta^{13}$C = +1.95 ‰) (Coplen, 1988). The mean external reproducibility (1σ) of carbonate standards is ± 0.05 ‰ for $\delta^{18}$O and ± 0.03 ‰ for $\delta^{13}$C. Measured NBS-18 $\delta^{18}$O is -23.2 ± 0.2
‰ VPDB and $\delta^{13}$C is -5.0 ± 0.1 ‰ VPDB. $\delta^{13}$C measurements were done at the highest possible resolution, depending on the availability of C. *wuellerstorfi* (usually every 1 to 2 cm).

### 2.1.2 Sedimentary Pa/Th

In contrast to *C. wuellerstorfi* $\delta^{13}$C, which reflects the nutrient content of bottom waters, sedimentary Pa/Th is a relatively recent tracer that records the renewal rate of water masses occupying the first ~1000 m above the seafloor (Thomas et al.,
2006). This tracer has been successfully used to reconstruct past changes in deep Atlantic circulation intensity (Gherardi et al., 2005, 2009; Guihou et al., 2010, 2011; Hall et al., 2006; Lippold et al., 2011, 2012; McManus et al., 2004).

$^{231}$Pa and $^{230}$Th are produced at a constant Pa/Th activity ratio of 0.093 by dissolved uranium, which is homogeneously distributed in the oceans. $^{230}$Th is however much more particle reactive than $^{231}$Pa, as reflected by their respective residence time in the ocean (30-40 y for $^{230}$Th, 200 y for $^{231}$Pa, Francois, 2007). $^{230}$Th is therefore rapidly removed from the water column
to the underlying sediment, while $^{231}$Pa can be advected by oceanic currents. High (low) flow rates therefore result in high



(low) $^{231}$Pa export and hence low (high) sedimentary Pa/Th ratio in the Atlantic. However, affinities of $^{231}$Pa and $^{230}$Th for settling particles depend on the particle type (Chase et al., 2002). For instance, $^{231}$Pa has a high affinity for opal, so that high opal fluxes can result in high sedimentary Pa/Th values even in the presence of lateral advection (Chase et al., 2002). The origin of sedimentary Pa/Th variability therefore needs to be carefully assessed.

5 Pa/Th measurements on core MD09-3256Q were performed by isotopic dilution mass spectrometry on a Thermo Finnigan MC-ICP-MS Neptune, following the method of Guihou et al., 2010.

Core SU90-03 sedimentary Pa/Th was measured by isotopic dilution on a single collector, sector field ICP-MS (Element2) at the University of British Columbia, following the procedure described by Choi et al., 2001.

For both cores, Pa and Th are corrected from radioactive decay since the time of sediment deposition and from authigenic and 10 lithogenic components using a $^{238}$U/$^{232}$Th ratio of 0.5±0.1 (Fig. S1) (Guihou et al., 2010).

### 2.1.3 Age model

Over the period 0-34 ka, core MD09-3256Q age model is based on 11 $^{14}$C dates measured on planktic foraminifer *G. ruber* white and converted to calendar age using the Marine13 curve with no additional reservoir age correction (Reimer et al., 2013) (Fig. S2). During the last glacial, Heinrich Stadials were recorded in marine sediment cores from the Brazilian margin as Ti/Ca 15 peaks resulting from increased terrigenous input during periods of increased precipitation associated with southward shifts in the position of the Inter Tropical Convergence Zone (ITCZ) (Jaeschke et al., 2007). Ti/Ca peaks are therefore good stratigraphic markers for correlating sediment cores with neighbouring well-dated cores. Therefore, from 34 ka to 50 ka, core MD09-3256Q was dated by correlation of its Ti/Ca record with that of core GeoB3910 using two tie points corresponding to the Ti/Ca peaks associated with HS4 and -5. GeoB3910 age model over the 34 to 50 ka period is based on one $^{14}$C date 20 calibrated using the Marine13 curve and on four speleothem tie points at the onset and end of HS4 and 5 (Burckel et al., 2015). Core MD09-3256Q age model and sedimentation rates are given in Table S2 and Fig. S3. The age model of core SU90-03 is based on 17 $^{14}$C dates measured on various species of planktic foraminifera (Chapman et al., 2000) that were converted to calendar ages using Marine13 calibration curve with no additional reservoir age correction.

$^{14}$C dates of all the published Atlantic cores used in this study were converted into calendar ages using the same method (Table 25 S1).

### 2.2 Circulation and Pa/Th model

### 2.2.1 Description of the models

In order to assess the vertical layout and renewal rate of the water masses constituting the AMOC during the last glacial period, the sedimentary Pa/Th data of the studied sediment cores were compared to Pa/Th values simulated with a simple 2D box 30 model (Luo et al., 2010) forced by different streamfunctions (Fig. 2b, d, f). Streamfunctions were generated using the iLOVECLIM coupled climate model, comprising atmosphere, ocean and vegetation components (Roche et al., 2014). A LGM equilibrium state computed using the PMIP-2 protocol was used as background climate (see Roche et al., 2007) for details). Streamfunctions used to mimic HS1 and a complete shutdown of the AMOC (off-mode) were generated by imposing a 0.16 and 0.35 Sv freshwater forcing in the Labrador Sea, respectively (Roche et al., 2010, 2014). The freshwater input is added 35 during 300 years on the LGM background state. The streamfunctions are taken as the mean over the 100-year period of lowest deep-water formation in the North Atlantic, during or right after the period of freshwater forcing. A freshwater input of 0.16 Sv allows the presence of a shallow circulation cell in the Atlantic Ocean, while a freshwater forcing of 0.35 Sv leads to an almost complete shutdown of the AMOC (Roche et al., 2014). Note that the freshwater input values needed to modify the AMOC are strongly model dependent and the important information carried by the model in the present context is the state of




the AMOC rather than the freshwater input value. Contrary to HS1 and off-mode streamfunctions, the Holocene streamfunction was computed using data based geostrophic velocity estimates (Talley et al., 2003).

Dissolved Pa and Th concentrations in the 2D box model are controlled by (1) production from U decay in the water column, (2) adsorption and desorption on settling particles and (3) advection by oceanic circulation. Particulate Pa and Th concentrations are controlled by (1) adsorption and desorption from the dissolved pool and (2) removal of sedimentary particles to the seafloor (Luo et al., 2010).

Modelled sedimentary Pa/Th meridional sections generated with the different streamfunctions are shown in Fig. 2a, c, e. Different water mass configurations result in different simulated sedimentary Pa/Th (Luo et al., 2010). In the deep Atlantic, increasing circulation intensity above a specific location causes Pa/Th to decrease at that water depth because of the increased Pa export and conversely. Increasing water depth without modifying circulation intensity also causes sedimentary Pa/Th ratio to decrease in the model because of the increased residence time of Pa and resulting higher Pa export, and conversely. Finally, the sedimentary Pa/Th ratio increases along the flow path of any water mass as low dissolved Pa concentrations of newly formed water masses increase by desorption of Pa from Pa-concentrated settling particles equilibrating with ambient waters (Francois, 2007). Adsorption and desorption rate constants also impact the simulated Pa/Th ratio. These constants were adjusted to reflect the opal belt in the southern ocean (Luo et al., 2010). For the Holocene, these constants were also changed to reflect opal in the northern North Atlantic (Lippold et al., 2012).

### 2.2.2 Limits of the models

The Pa/Th model is a 2D model without parameterization of diffusive transport (Luo et al., 2010). This prevents the model from simulating boundary scavenging, which is the transfer of dissolved protactinium from open ocean regions of high Pa concentrations to coastal regions of low Pa concentration such as in upwelling zones (Christl et al., 2010). However, as described in the results section, we verified that our Pa/Th signal is mainly driven by oceanic circulation changes and the importance of diffusive transport is therefore likely negligible here. This simple 2D Pa/Th model therefore appears adequate for comparison with our Pa/Th data.

The vertical resolution of the iLOVECLIM model is depth dependent, with higher resolution (10 to 100 m) in the upper water column than below 1000 m (500 to 700 m). Hence, the uncertainty in the position of the water mass transitions in the streamfunctions below 1000 m is of 500 to 700 m. However, because sedimentary Pa/Th likely reflects the protactinium export in the bottom 1000 m of the water column (Thomas et al., 2006), the model vertical resolution is sufficient to properly simulate sedimentary Pa/Th values. Moreover, benthic foraminiferal $\delta^{13}C$ measurements, which reflect the DIC of the water mass directly above the sediment interface, allows confirming or infirming the geometry information contained in measured Pa/Th values. Hence, the relatively low vertical resolution of the iLOVECLIM model in the deep-ocean does not affect our conclusions.

### 2.3 Time slice definition

We define three interstadial and one Heinrich Stadial time slices (Fig. 3) to compare Pa/Th data measured in Atlantic cores to Pa/Th values simulated with the different streamfunctions. We focus on HS2, the preceding GI and the GIs bracketing HS4. We did not include HS4 in our study because core MD09-3257 HS4 Pa/Th data are affected by boundary scavenging (see Sect. 3.1, Burckel et al., 2015) and we therefore lack information in an important location of the Atlantic Ocean.

GI-3, GI-8 and GI-10 time slices are defined as the periods of stable sedimentary Pa/Th values in core MD09-3257 associated with the NGRIP GI time intervals (Fig. 3). More specifically, we used as a reference MD09-3257 Pa/Th values bracketing the middle of NGRIP GI time intervals in the GICC05 age scale (Rasmussen et al., 2014). Contiguous Pa/Th values within 1 sigma uncertainty of the Pa/Th reference value form a plateau of stable Pa/Th values that was used to define the GI time slices. With



this definition, GI time slices represent the periods of stable oceanic circulation associated with each GI. The HS2 time slice was defined in core MD09-3257 as the period of maximum sedimentary Pa/Th after the abrupt rise associated with the onset of HS2.

Sedimentary Pa/Th values associated with each time slice and core are given in Table S3. We computed uncertainties on the
Pa/Th values associated with each time slice accounting for the uncertainty on individual Pa/Th measurements and uncertainties on the age model (see Supplementary Information).

Both GI-8 and -10 time slices are associated with high temperatures recorded in Greenland ice cores. The GI-3 time slice includes both the period of high Greenland temperatures associated with GI-3, and periods of low temperatures associated with GS-4 and the beginning of GS-3. Given the low Pa/Th and high $\delta^{13}C$ values in the intermediate equatorial core at that
time, we consider that the GI-3 time slice mainly reflects interstadial conditions. However, because temporal resolution of the marine records is too low to clearly distinguish between GI-3 and GS-4, information about the state of the AMOC during GIs derived from this time slice should be considered with caution.

### 2.4 Quantification of the model-data agreement

In order to quantify the agreement between simulated and measured sedimentary Pa/Th, we compute the Euclidean distance,
defined as the square root of the sum of squared differences between simulated and measured Pa/Th for each core. Minimum values indicate the best agreement between simulated and measured Pa/Th (Tables S4 and S5).

### 3 Results

### 3.1 Sedimentary Pa/Th data

Cores MD09-3257, MD09-3256Q and SU90-03 Pa/Th measurements were centred on HS2 and HS4 (Fig. 3). No Pa/Th values
were measured within HS2 in core MD09-3257 because of the presence of turbidite layers (Burckel et al., 2015). During HS4 and before HS2, the sedimentary Pa/Th ratio of core MD09-3257 rises above the production ratio of 0.093, indicating the absence of Pa export. Pa/Th variability associated with GS and GI is observed, with high Pa/Th values occurring during GS and low Pa/Th values during GI. Pa/Th variations in core MD09-3256Q are more muted (Fig. 3). The main Pa/Th variation in core MD09-3256Q occurs during HS4, when Pa/Th values rise from ~0.06 to ~0.08. This increase in MD09-3256Q Pa/Th also
corresponds, within dating uncertainties, with the largest Pa/Th change from ~0.03 to ~0.05 in core SU90-03.

Before interpreting our Pa/Th records in terms of ocean circulation changes, we need to assess whether varying lithogenic or opal fluxes impacted the scavenging intensities of Pa and Th. To do so, we use the preserved opal and $^{232}Th$ fluxes as tracers for past opal and terrigenous fluxes respectively (Anderson et al., 2006; Lippold et al., 2012). Core MD09-3257 Pa/Th data are mainly influenced by oceanic circulation, except during the high lithogenic flux period associated with HS4 ($^{232}Th$ flux >
12 dpm.cm$^{-2}$.ky$^{-1}$, Fig. 3a, white squares) (Burckel et al., 2015). In core MD09-3256Q, opal fluxes do not covary with the Pa/Th ratio and are very low (0.01-0.02 g.cm$^{-2}$.kyr$^{-1}$, Table S6) (Fig. S4a). In the Atlantic, the lowest opal flux value observed to influence the sedimentary Pa/Th ratio is 0.2 g.cm$^{-2}$.kyr$^{-1}$ (Lippold et al., 2012). Hence, given the much lower opal fluxes recorded in core MD09-3256Q and their lack of correlation with sedimentary Pa/Th, we conclude that biogenic silica had no or very little influence on Pa/Th variability. Similarly, we find no correlation between $^{232}Th$ fluxes and Pa/Th values in this
core (P value = 0.48, n = 22) (Fig. S5a). We can therefore safely assume that the Pa/Th variability recorded in our equatorial cores is mainly driven by changes in oceanic circulation intensity.

Core SU90-03 opal fluxes are low (< 0.1 g.cm$^{-2}$.kyr$^{-1}$, Table S6) and do not show any correlation with Pa/Th data (P value = 0.52, n = 10) (Fig. S4b). In contrast, $^{232}Th$ fluxes could be correlated to the Pa/Th signal (P value = 0.03, n = 16, Fig. S5b). This correlation is only driven by the highest Pa/Th value, and removing this single value results in the disappearance of the



correlation (P value = 0.31, n = 15). However, we chose to keep this value as SU90-03 $^{232}$Th flux is low (< ~1.5 dpm.cm$^{-2}$.kyr$^{-1}$) and its Pa/Th signal remains low (0.03-0.05) on the entire 20-50 ka period, indicating a constant significant Pa export through water mass advection. Oceanic circulation is therefore the main process explaining SU90-03 Pa/Th data.

The published Pa/Th values of other sediment cores used in this study have been shown to be mainly driven by oceanic

circulation intensity (Bradtmiller et al., 2014).

### 3.2 C. wuellerstorfi δ13C data

*C. wuellerstorfi* $\delta^{13}$C of both equatorial cores shows millennial-scale variability, with low $\delta^{13}$C values occurring during HS. GS and GI are also recorded in the $\delta^{13}$C record of the intermediate core by low and high $\delta^{13}$C values, respectively. In the deep core, the low sedimentation rate induces a low temporal resolution and a smoothing of the $\delta^{13}$C signal that may have erased

$\delta^{13}$C decreases associated with short GS.

$\delta^{13}$C values during the LGM are 0.24 ± 0.07 ‰ and 0.66 ± 0.06 ‰ in the deep and intermediate equatorial core, respectively (Fig. 3). Given that the late Holocene $\delta^{13}$C measured in the equatorial cores are both close to 1.35 ‰ (see supplementary information), these LGM $\delta^{13}$C values are much lower than what would be expected from the ~0.3‰ glacial-interglacial change in mean ocean $\delta^{13}$C in response a reduced continental biosphere during glacial periods (Duplessy et al., 1988). The low $\delta^{13}$C

values observed during the LGM and some of the GS could thus indicate either a slowdown of the deep-water circulation, or an increased influence of southern sourced water masses at both sites during the glacial with respect to the Holocene.

### 3.3 Ocean circulation signals

In cores MD09-3257/GeoB3910, SU90-03 and ODP1063, GI time slices are generally characterized by lower Pa/Th and higher $\delta^{13}$C values than those characterizing GS periods (Fig. 3). The only exception is GI-10 as this period is associated with a

transition from relatively low to high Pa/Th values associated with HS4 in core SU90-03. Conversely, the HS2 time slice is associated with high Pa/Th values and low $\delta^{13}$C values in these cores, except for core SU90-03 that exhibits Pa/Th and $\delta^{13}$C values similar to GI time slices.

Core MD09-3256Q Pa/Th values are below 0.07 in all the studied time slices, with minor variability even between GI and HS2 time slices. Its $\delta^{13}$C record is systematically 0.4-0.5 ‰ below that of core MD09-3257, even though the $\delta^{13}$C difference

between the two cores is reduced during HS2. Core MD09-3256Q therefore appears to be constantly bathed by a nutrient rich water mass exporting dissolved protactinium.

There is only one Pa/Th data within our GI time slices in core MD02-2594 (GI-3, Fig. 3). However, two other Pa/Th values can be attributed to both GI-8 and GI-10 as they lie within these time slices considering age model uncertainties. These low MD02-2594 Pa/Th values indicate that, during GI time slices, protactinium was exported away from the intermediate-depth

South Atlantic Ocean.

In addition to the above Pa/Th records, we use one Pa/Th value from core V29-172. This value lies within the GI-3 time slice and is rather low (0.04), similarly to the shallower North Atlantic core SU90-03. This supports the existence of Pa export between 1500 and 3500 m depth in the North Atlantic during GI-3.

### 4 Discussion

Time slice sedimentary Pa/Th data from the six selected Atlantic sediment cores are compared, when available, to the sedimentary Pa/Th pattern simulated in response to the different streamfunctions (Fig. 4-6). We focus in particular on the equatorial cores and describe their modelled and measured sedimentary Pa/Th by referring to the vertical Pa/Th gradient between 2300 and 3500 m (i.e. the water depths of the sediment cores). Indeed, vertical Pa/Th gradients are useful indicators of the vertical layout of water masses. In what follows, we will see how each vertical gradient can be interpreted.



### 4.1 Greenland interstadials (GI-3, GI-8 and GI-10 time slices)

#### 4.1.1 Comparison with the Holocene simulation

A large vertical Pa/Th gradient between the two equatorial core sites is simulated by the model in the presence of a southward flowing northern-sourced deep water mass such as NADW in the Holocene simulation (Fig. 4). This is due to the fact that the

sedimentary Pa/Th ratio decreases with depth within a single water mass of uniform flow rate. This effect is intensified in the case of the Holocene streamfunction, as the flow rate of NADW is not uniform but stronger between 2500-3500 m and weaker between 1300-2300 m (acquisition depths of sedimentary Pa/Th for the deep and intermediate cores respectively). Hence, Pa export at 3500 m is more intense than at 2300 m, thereby significantly increasing the vertical Pa/Th gradient between the two cores.

Interstadial Pa/Th data in the deep equatorial and North Atlantic cores are consistent with simulated Pa/Th values obtained with the Holocene streamfunction (Fig. 4b-d). However, interstadial sedimentary Pa/Th values in the equatorial core at intermediate depth are lower than predicted by the Pa/Th model forced with the Holocene streamfunction. The vertical Pa/Th gradient between our equatorial cores during interstadials is small, which is in contradiction with the large vertical Pa/Th gradient simulated with the Holocene streamfunction. Moreover, data from Southern Ocean core MD02-2594 are

systematically between 0.045 and 0.050 during GI (Fig. 3), and in conflict with the high Holocene Pa/Th value (~0.09) simulated at this core site (Fig. 4).

High $\delta^{13}$C values in the intermediate equatorial core during MIS3 interstadials suggest that northern-sourced deep waters influenced the equatorial Atlantic at 2300 m depth (Fig. 3b). However, the lower $\delta^{13}$C values of the deep equatorial core imply that, unlike in the present-day Atlantic, nutrient-rich southern-sourced deep waters were present at 3500 m depth in the

equatorial West Atlantic.

Therefore, both Pa/Th and $\delta^{13}$C data indicate that the geometry and strength of the AMOC during the studied GI were different from those of the Holocene.

#### 4.1.2 Comparison with the off-mode simulation

Pa/Th values simulated with the off-mode streamfunction exhibit a small vertical Pa/Th gradient between the two equatorial

cores (Fig. 5). However, the sedimentary Pa/Th values measured during interstadials are much lower than the simulated values at both equatorial sites and in the North Atlantic Ocean. These low measured Pa/Th values imply a significant export of Pa by oceanic circulation and therefore exclude the possibility of an almost halted deep Atlantic circulation above 3500 m depth (Fig5, b-d).

In addition, while in the off-mode streamfunction no significant deep convection occurs in the high-latitude North Atlantic,

the high $\delta^{13}$C values of core SU90-03 and MD09-3257 indicate that northern sourced waters were present at ~2500 m in the North and equatorial Atlantic (Fig. 3).

Hence, it is highly unlikely that the off-mode streamfunction depicts the deep Atlantic circulation during the studied GI.

#### 4.1.3 Comparison with the HS1 simulation

The HS1 streamfunction also induces a small vertical Pa/Th gradient between the equatorial core locations (Fig. 6). In contrast

to the simulation obtained with the off-mode streamfunction, this small vertical Pa/Th gradient is associated with significant lateral export of Pa at the depth of both the intermediate and deep equatorial cores, as well as at the Bermuda Rise and Southern Ocean cores, in agreement with Pa/Th data (Fig. 6b-d). Such a small vertical Pa/Th gradient is simulated in the case of two water masses overlying each other and flowing in opposite directions (Fig. S6) (Lippold et al., 2012). Indeed, in the HS1 streamfunction, northern-sourced waters affect the depth of the intermediate equatorial core (above 2500 m). Below ~4000 m,

northward flowing southern-sourced waters are active and lead to a return flow (between 2500-4000 m depth) that influences



the depth of the deep equatorial core (3500 m) (Fig. 2e, f). This circulation scheme results in decreasing or invariant lateral export of Pa with depth, which in turns causes sedimentary Pa/Th to increase or to be constant with depth at the equator.

However, the agreement between simulated and measured Pa/Th in the Atlantic Ocean cores is lower (i.e. larger Euclidean distances, Table S4) with the HS1 streamfunction than with the Holocene streamfunction. The better model-data agreement

obtained with the Holocene streamfunction is driven by the Pa/Th data of the intermediate and deep mid-latitude North Atlantic cores (SU90-03 and V29-172), as the deep convection of NADW induces low modelled Pa/Th values south of 50°N. In the HS1 streamfunction, the region of deep-water formation is shifted southward (Fig. 2). In the Pa/Th model, dissolved protactinium and thorium concentrations are therefore vertically homogenized between 40 and 60°N (against 60-70°N in the case of the Holocene streamfunction), preventing the presence of low sedimentary Pa/Th values north of ~40°N. A southward

shift in the deep convection zone during the last glacial has been observed in earlier studies (e.g. Vidal et al., 1997), but the simulated southward shift in the present HS1 streamfunction could be overestimated. Assuming a more northerly position of the region of deep-water formation, the Pa/Th observed in the North Atlantic cores would agree with the HS1 streamfunction. Indeed, if we remove both mid-latitude North Atlantic cores from the computation of the sum of squared residuals, we find that the HS1 streamfunction best explains the Pa/Th data observed during GI (Table S5).

Interstadial benthic $\delta^{13}C$ values at the equator indicate the presence of (1) a northern-sourced water mass at the intermediate core site and (2) of a southern-sourced water mass at the deep core site (Fig. 3b). Hence, benthic $\delta^{13}C$ data support the existence of two water masses overlying each other and flowing in opposite directions as in the HS1 streamfunction. Moreover, the southern sourced water mass likely affected core MD02-2594 on its way towards the deep equator site as reflected by its low benthic $\delta^{13}C$ (< 0.5 ‰, Negre et al., 2010).

There are several studies discussing a potential link between D-O events and Atlantic circulation changes (Gottschalk et al., 2015: see Boyle, 2000 for a review), but to our knowledge, there has been no study about the geometry of the AMOC water masses during these periods. Combining the information provided by sedimentary Pa/Th and benthic foraminiferal $\delta^{13}C$ data, we reach the following conclusions concerning the Atlantic circulation below ~1300 m during the studied GI. A southward-flowing northern-sourced water mass likely circulated above ~2500 m, while southern-sourced deep water circulated

northwards below ~4000 m, and southwards as a return flow between ~2500 and 4000 m depth (Fig. 2f, Fig. 7a). Moreover, our data indicate that the geometry and state of the AMOC appear similar for GI-3, -8 and -10, despite the different ice sheet volumes characterizing the periods encompassing HS2 and HS4 respectively.

### 4.1.4 Estimation of the AMOC intensity over the interstadial time slices

Our Holocene equatorial Pa/Th values are in reasonable agreement with previously published data from the Brazilian margin

(Lippold et al., 2011) and with Pa/Th values simulated with a 2 and 3 fold increased Holocene streamfunction (Fig. S7). At present, increasing the Holocene streamfunction is indeed necessary to improve the agreement between simulated and measured equatorial Pa/Th values (Lippold et al., 2011). This increase was proposed to account for the absence of west-east difference in circulation strength in the 2D Pa/Th model, which reflects a zonally averaged circulation. Moreover, the width of the Atlantic basin is the shortest at the equator, while it is assumed constant in the model. Both these effects could cause the

flow speed at the equator to be underestimated, and therefore the simulated Pa/Th ratio to be overestimated.

We performed sensitivity tests of the Pa/Th model to varying flow rates by multiplying the HS1 streamfunction by a factor of 1, 2 and 3. Our results indicate that factors of 1 and 2 best reproduce the sedimentary Pa/Th ratio in the Atlantic during GI time slices (Table S5).

When using the HS1 streamfunction, the flow of both the northern and southern sourced water masses at the equator is of 5-

10 Sv. As this streamfunction amplified by a factor 1 and 2 best agrees with our interstadial Pa/Th data, we assume that the water-mass flow rates provided by these HS1 streamfunctions (5-20 Sv) depict well the oceanic circulation strength during the studied interstadials. The modern flow rates of NADW (northern-sourced water mass) and AABW (southern-sourced water



mass) are ~27 ± 7 and ~3 Sv at 4.5°S respectively (Lux et al., 2001). We conclude that during GI-3, -8 and -10, the flow rate of the southern-sourced deep water was likely larger than present day AABW, and that the flow rate of the northern-sourced deep water may have been smaller than present-day NADW.

**4.2 Heinrich Stadial 2**

**4.2.1 Comparison with the HS1 simulation**

Pa/Th data from the HS2 time slice display a large vertical Pa/Th gradient between the depths of our equatorial cores (Fig. 4a, 5a, or 6a). The gradient results from the high Pa/Th value at intermediate depth, which indicates that the core was likely overlain by sluggish waters. Pa/Th data in the equatorial cores are therefore incompatible with the low vertical Pa/Th gradient simulated by the HS1 streamfunction (Fig. 6, Table S4).

Moreover, the decrease in benthic $\delta^{13}$C values in both equatorial cores during HS2 suggests an increased influence of southern-sourced deep waters in the deep Atlantic at both 3500 and 2300 m, which is incompatible with the active northern sourced circulation cell simulated above 2500 m with the HS1 streamfunction (Fig. 3, b).

Hence, both circulation and ventilation proxies indicate that there was no intense northern sourced water flow between 1300 and 2300 m depth at the equator during HS2.

**4.2.2 Comparison with the Holocene simulation**

Large vertical Pa/Th gradients are simulated in the model when a single water mass affects both equatorial cores. Pa/Th values simulated with the Holocene streamfunction therefore best fit Pa/Th data during HS2 (Fig. 4a, Table S4). However, the very low benthic $\delta^{13}$C values measured in both equatorial cores during HS (Fig. 3b) exclude the presence of an active northern sourced deep-water mass in the intermediate and deep equatorial Atlantic at that time.

Hence, it is unlikely that the Holocene streamfunction depicts the geometry and strength of the AMOC during HS2.

**4.2.3 Comparison with the off-mode simulation**

In contrast, Pa/Th values simulated with the off-mode streamfunction could reconcile the large Pa/Th vertical gradient and the low $\delta^{13}$C of equatorial cores (Fig. 5a). In the current off-mode simulation by the iLOVECLIM model, the vertical extent of the southern-sourced water mass is not large enough to influence the equatorial core at 3500 m depth, resulting in an apparent low 25 vertical Pa/Th gradient (Fig. 2c, d). However, the model was run for a short period of time, preventing a full response of the southern sourced deep waters to varying climatic and oceanographic conditions (Roche et al., 2014). Hence, the vertical extent of the southern sourced water mass could be larger, thereby inducing a low sedimentary Pa/Th ratio in the deepest core of the Brazilian margin. Pa/Th data from the Bermuda Rise and South Atlantic cores are consistent with simulated Pa/Th values from the off-mode streamfunction. However, the Pa/Th data from the intermediate North Atlantic core is not in agreement with the 30 modelled Pa/Th, as there is no deep-water formation in the high latitude North Atlantic in the off-mode streamfunction.

The very low benthic $\delta^{13}$C values measured in the deep equatorial core during HS2 is consistent with the simulation using the off-mode streamfunction, which shows a strong influence of southern sourced water masses in the deep Atlantic. This is further supported by the increased $\varepsilon_{Nd}$ in the Bermuda Rise core which indicates an increased influence of southern-sourced water masses at 4500 m in the North Atlantic Ocean during HS2 (Gutjahr and Lippold, 2011).

The off-mode simulation could therefore explain both Pa/Th and $\delta^{13}$C values in the deep Atlantic and intermediate equatorial Atlantic, but neither this streamfunction nor the others are able to explain the Pa/Th record of the North Atlantic core at ~2500 m depth.





### 4.2.4 Geometry and strength of the AMOC at the onset of HS2

The low Pa/Th and high benthic $\delta^{13}C$ observed in core SU90-03 indicates that deep convection was still active in the high latitude North Atlantic during HS2. However, the resulting water mass was probably slow enough so that its nutrient content significantly raised on its way towards the equator either through turbulent and diffusive mixing with the underlying water mass or through the degradation of $^{12}C$ rich organic matter sinking from surface waters. This would explain the low $\delta^{13}C$ values measured in the intermediate equatorial core. Similarly, a long transit time of the water mass from the North Atlantic deep convection sites to the Brazilian margin could explain the high Pa/Th values measured in the intermediate equatorial core, as dissolved Pa was allowed sufficient time to equilibrate through reversible exchange with particulate matter. This progressive equilibration would have led to the sedimentary Pa/Th latitudinal gradient observed at ~2500 m depth, from low sedimentary Pa/Th values in the mid-latitude North Atlantic (SU90-03) to high values in the equatorial Atlantic (MD09-3257) (Fig. 5a). Considering that both Pa/Th and benthic $\delta^{13}C$ values suggest an increased influence of southern sourced waters at the equatorial and Bermuda Rise deep sites, the off-mode streamfunction likely best depicts the geometry of the AMOC during HS2 below 2500 m, i.e. southern-sourced deep waters likely dominated the deep Atlantic Ocean (Fig. 7b). The direct influence of the southern-sourced water mass likely extended vertically above 3500 m depth, probably at least up to 2500 m depth, as indicated by the low Pa/Th value of the deep equatorial core. However, given the discrepancy between the off-mode simulation and observed Pa/Th values in the North Atlantic intermediate core, it is difficult to assess the exact position of the southern sourced waters and their associated return flow. Above 2500 m, a weak water flow originating from Northern deep convection sites likely influenced the Atlantic, perhaps down to equatorial latitudes (Fig. 7b).

The geometry of the Atlantic deep water masses inferred from our sedimentary Pa/Th and benthic $\delta^{13}C$ records during HS2 is consistent with previous modelling experiments (e.g. Ganopolski and Rahmstorf, 2001) and with water mass ventilation (Elliot et al., 2002; Vidal et al., 1997; Zahn et al., 1997) and circulation intensity (Gherardi et al., 2005; McManus et al., 2004) proxies indicating that deep water circulation slowed down during Heinrich Stadials. Furthermore, our data indicate that a northern-sourced water mass was active above 2500 m during HS2, as inferred for HS1 (Gherardi et al., 2009; Roche et al., 2014) and HS2 (Lynch-Stieglitz et al., 2014; Wary et al., 2015) in previous studies. However, quantifying the intensity of the AMOC upper circulation cell during this period remains difficult since there is at present no numerical simulation in reasonably good agreement with both the circulation and ventilation proxies measured in the Atlantic.

### 5 Conclusions

We have shown that both the geometry and strength of the AMOC during three interstadials of the last glacial period (i.e. the GI-3, GI-8 and GI-10 intervals) was markedly different from those of the modern AMOC. Our data suggest that a northern-sourced water mass circulated above 2500 m depth with a flow rate ranging between 5 and 20 Sv, which is lower than the intensity of present-day NADW. Below 4000 m, a southern-sourced deep water mass likely flowed northward with an intensity of 5-20 Sv, which is larger than the modern AABW flow rate. Between 2500 and 4000 m depth, the southern-sourced deep water likely circulated southwards as a return flow. Our data show that the geometry of the AMOC changed at the onset of HS2 and that the deep Atlantic below 2500 m was probably dominated by a single southern-sourced water mass that can be traced up to 35°N at 4500 m. This water mass probably directly affected the equatorial Atlantic between 2500 and 3500 m depth. A slow southward flowing water likely circulated between 1500 and 2500 m in the North Atlantic, but its presence at the equator remains difficult to assess.





**Author contribution**

P. Burckel, C. Waelbroeck, S. Pichat, J. Gherardi and J. Lippold designed the research, P. Burckel, S. L. Jaccard, and F. Thil performed the sedimentary Pa/Th measurements, S. L. Jaccard and J. Lippold performed the opal measurements, A. Govin performed the XRF measurements, Y. Luo generated the simulated Pa/Th data and D. Roche generated the streamfunctions.

P. Burckel and C. Waelbroeck wrote the manuscript.

**Data availability**

Data related to this article are available as Supplementary Information files.

**Acknowledgments**

Pa/Th measurements were funded by the CNRS/INSU LEFE project ACCENT. This work is a contribution to the RETRO
project, a joint European Science Foundation (ESF)/EUROMARC, funded by Research Council of Norway (RCN), France (CNRS/INSU), Germany, and the Netherlands, and to the ACCLIMATE ERC project funded by the European Research Council under the European Union's Seventh Framework Programme (FP7/2007-2013)/ERC grant agreement no 339108. Cores MD09-3256Q and MD09-3257 were collected on board R/V Marion Dufresne during RETRO Cruise III, supported by ESF EUROMARC project RETRO, IPEV and ANR project ANR-09-BLAN-0347. We thank the IPEV team, crew members
of R/V Marion Dufresne and all scientists who participated in RETRO Cruise III. We are thankful to M. Roy-Barman for expert advices on Pa/Th measurements on LSCE MC-ICP-MS. We acknowledge C. Moreau, J.-P. Dumoulin, and the UMS ARTEMIS for AMS $^{14}$C dates, and L. Mauclair and F. Dewilde for invaluable technical assistance. This is LSCE contribution xxxx.

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

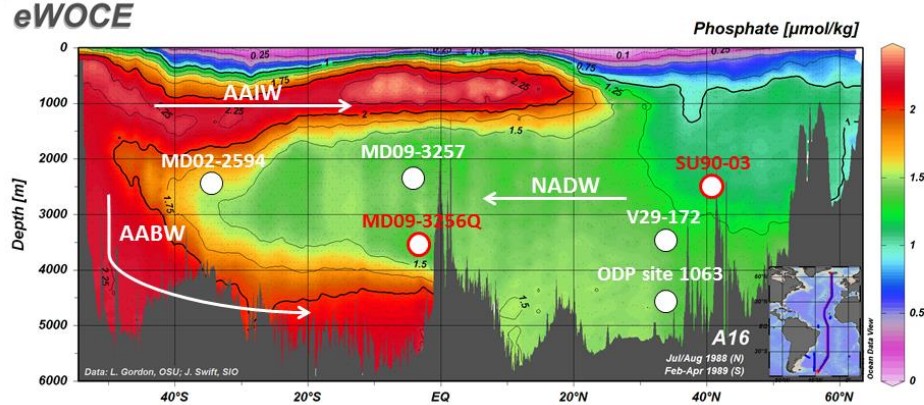

**Figure 1: Phosphate section of the Atlantic Ocean (Schlitzer, 2000) showing the location of the studied sediment cores (see Sect. 2.1 or Table S1 for detailed locations). Cores for which we provide new Pa/Th or δ¹³C data are circled in red. Phosphate content follows the structure of the present day AMOC. White arrows indicate the approximate flow directions of the Antarctic Intermediate Water (AAIW), Antarctic Bottom Water (AABW) and North Atlantic Deep Water (NADW).**

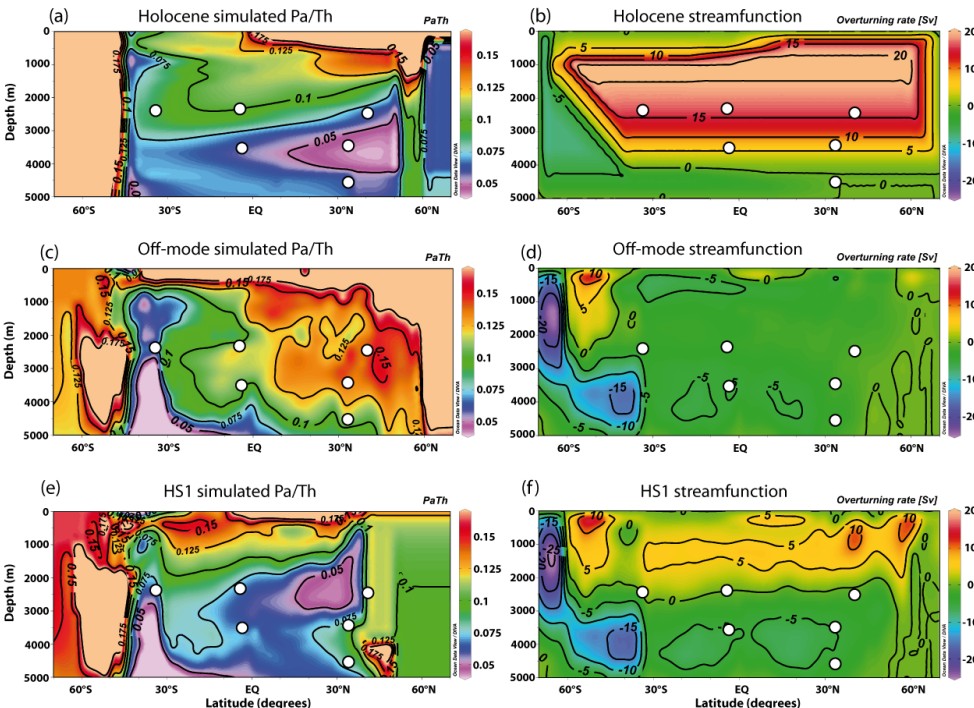

**Figure 2: Simulated sedimentary Pa/Th values (left) in response to different streamfunctions (right): (a, b) Holocene, (c, d) Off-mode, (e, f) HS1. White dots indicate the position of the studied sediment cores (see Fig. 1). Data gridding was achieved using the Ocean Data View software (Schlitzer, 2015).**



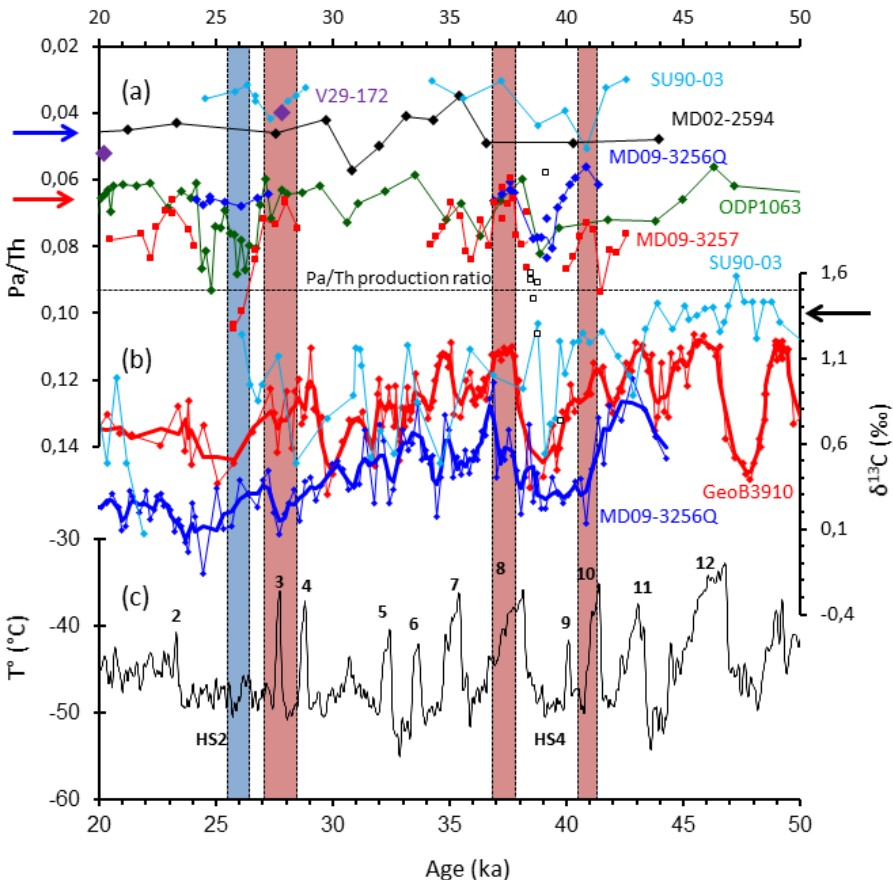

**Figure 3: Comparison between sedimentary Pa/Th and benthic δ¹³C data from the Brazilian margin, Bermuda Rise, mid-latitude North and South Atlantic Ocean and Greenland temperatures. (a) MD09-3256Q (this study), SU90-03 (this study), MD09-3257 (Burckel et al., 2015), ODP1063 (Böhm et al., 2015), MD02-2594 (Negre et al., 2010) and V29-172 (Bradtmiller et al., 2014) Pa/Th**

5   **(b) MD09-3256Q (this study), SU90-03 (Chapman et al., 2000) and GeoB3910 (Burckel et al., 2015) *Cibicides wuellerstorfi* δ¹³C and (c) NGRIP temperature record on the GICC05 timescale (Kindler et al., 2014). In (a) the average Pa/Th for each core is represented by the lines and individual measurements by diamonds or squares (MD09-3257). V29-172 Pa/Th values are represented by two purple diamonds. White squares indicate Pa/Th values not considered in core MD09-3257, as they might not be influenced by oceanic circulation only (Burckel et al., 2015). The red and blue arrows indicate the Late Holocene Pa/Th values in cores MD09-**

10   **3257 (0.065 ± 0.004, Burckel et al., 2015) and MD09-3256Q (0.043±0.002), respectively. Error bars on Pa/Th measurements are given in Fig. S1 and Table S7 and S8. In (b) thick lines are 3-point running averages of the *Cibicides w.* δ¹³C records, the black arrow indicates present day NADW δ¹³C value (~1.36 ‰, supplementary information). δ¹³C values are given in Table S9. In (c) numbers indicate the GI. Red vertical bands represent the GI-3, GI-8 and GI-10 time slices and the blue vertical band the HS2 time slice.**




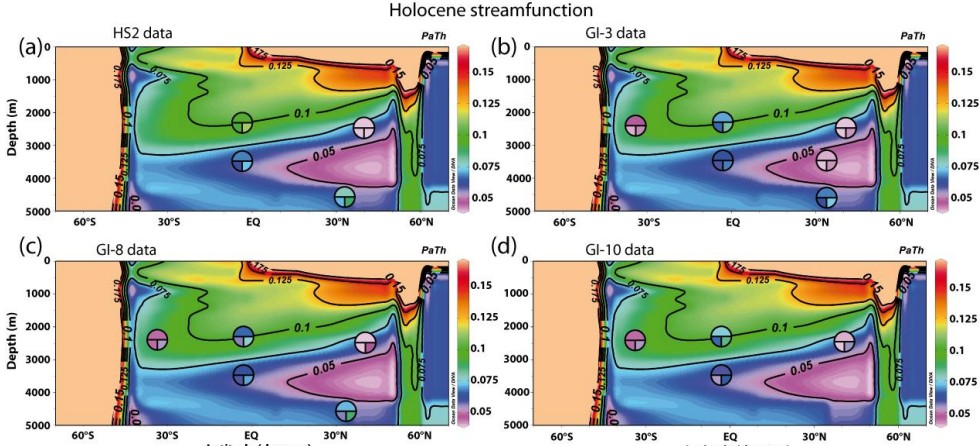

**Figure 4: Comparison of the Pa/Th data (circles) for each of the time slices to the simulated Pa/Th values using the Holocene streamfunction. (a) HS2, (b) GI-3, (c) GI-8 and (d) GI-10 Pa/Th data. The upper half of the circles represents the Pa/Th mean value, the lower left quarter, the Pa/Th mean value – 1 sigma, and the lower right quarter, the Pa/Th mean value + 1 sigma. Data gridding was achieved using the Ocean Data View software (Schlitzer, 2015).**

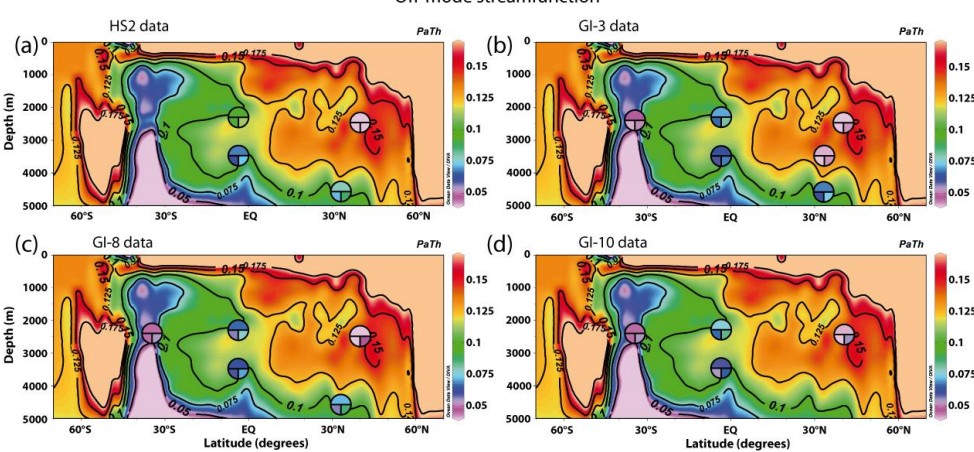

**Figure 5: Comparison of the Pa/Th data for each of the time slices to the simulated Pa/Th values using the off-mode streamfunction. Time slices are (a) HS2, (b) GI-3, (c) GI-8 and (d) GI-10 Pa/Th data; symbols as in Fig. 4.**



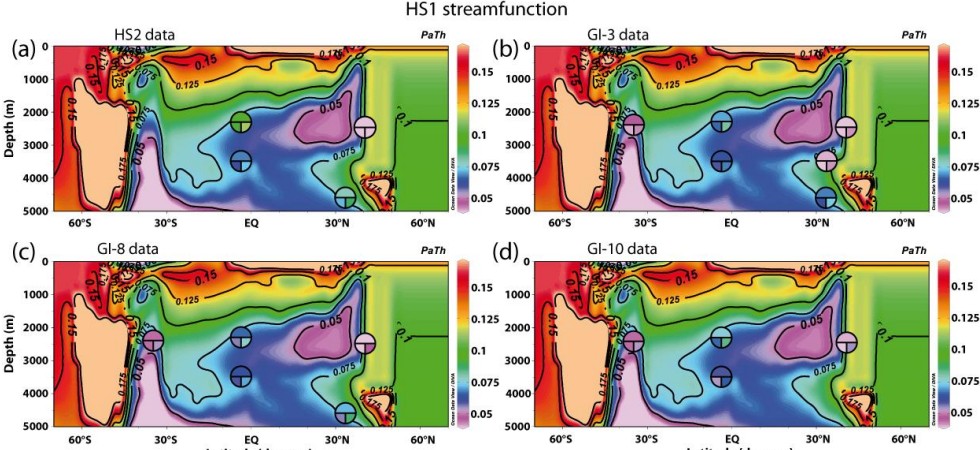

**Figure 6: Comparison of the Pa/Th data for each of the time slices to the simulated Pa/Th values using the HS1 streamfunction. Time slices are (a) HS2, (b) GI-3, (c) GI-8 and (d) GI-10 Pa/Th data; symbols as in Fig. 4.**

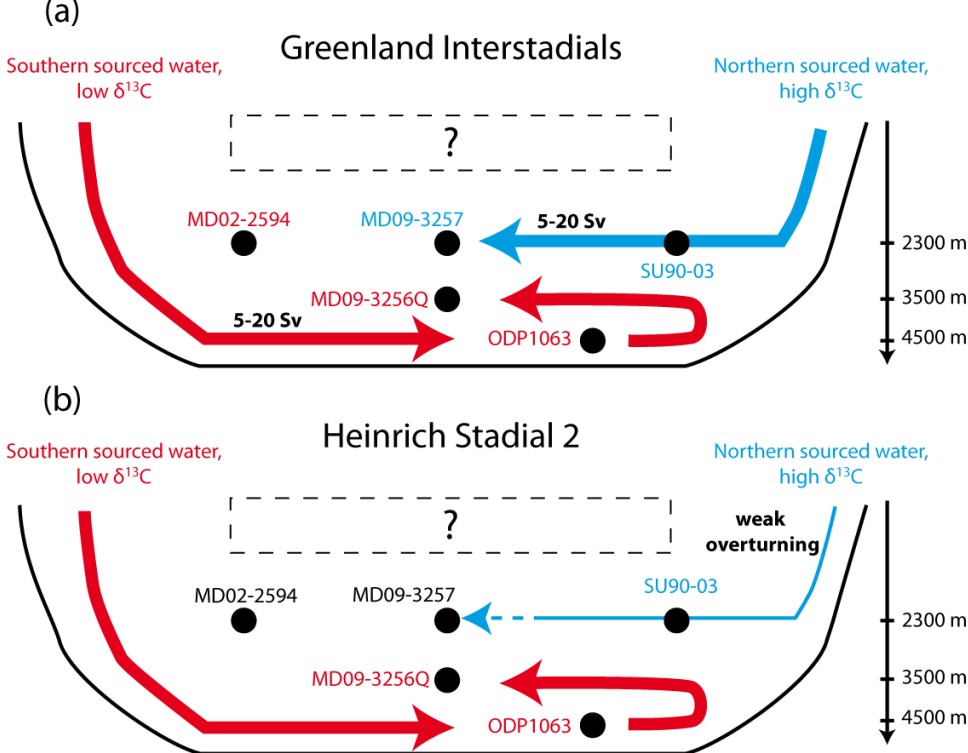

5   **Figure 7: Sketch of the possible states of the AMOC during (a) Greenland interstadials and (b) Heinrich Stadial 2. Red and blue arrows depict the southern and northern sourced water mass, respectively. The arrows' thickness reflects the overturning rate. Core names are coloured depending on which water mass influences them. In (b), cores MD09-3257 and MD02-2594 are written in black as it is difficult to assess which water mass bathes the equator at 2300 m depth and as we have no direct evidence of which water mass influences the South Atlantic Ocean during HS2. The exact position of the northward flowing southern sourced water mass**
10   **and its return flow is unknown in (b).**