# Peer review of "Changes in the geometry and strength of the Atlantic Meridional Overturning Circulation during the last glacial (20-50 ka)"

_Climate of the Past, 2016_

## Referee Comment (RC1) · Anonymous Referee #1 · 26 May 2016

In their manuscript "Changes in the geometry and strength of the Atlantic Meridional Overturning Circulation during the last glacial (20-50 ka)", Burckel et al. use 231Pa and 230Th ratios and 13C to assess the past state of deep ocean circulation in the Atlantic Ocean at several intervals during the past glaciation. After attempting to assess the geometry and strength of the overturning cell of the Atlantic, they conclude that the deep ocean circulation was very different from the modern in all four of their study intervals. The interstadial circulation was different in being relatively shallow, with a deep inflow from the south. Southward flowing waters at mid-depth would therefore have been the return flow of southern-sourced waters. At the time of Heinrich Stadial 2, yet another different circulation is inferred, with southern waters filling the deep Atlantic

and a slow, southward-flowing water mass occupying the intermediate depths.

This is a potentially valuable contribution to the literature on past states of the ocean circulation. It presents new geochemical data in a spatial array that may provide insights into changes at different depths and locations. The isotopic method is a promising and exciting approach, although it seems still in development in comparison to modern measurements. The data are compared to model output, which although limited in resolution and lacking a third dimension, nevertheless provides useful constraints on potential interpretations. The conclusions are not inconsistent with the relatively limited data presented.

In terms of the specific criteria, the paper certainly addresses relevant questions within the scope of CP. It does not present novel approaches, but builds well upon existing techniques, data, and ocean modeling output. Substantial conclusions are reached regarding the configuration and rate of ocean circulation. The conclusions are not inconsistent with the data, although there are too many gaps at relevant locations and depths for them to be any more convincing than many alternatives which are not discussed. Figures are relatively clear, and text is a reasonable length. The text is fluent and the authors give adequate credit to the previous studies that they utilize and discuss.

The two largest issues with the paper in its present form are related to its justification and chronology. This is a study of four time slices that are widely distributed within the last glacial. They are neither the most extreme, nor the most characteristic. Nor do they include important transitions or intervals of special climatic interest. It is therefore not clear to the reader why this seemingly arbitrary assortment of time slices was chosen. The authors should provide a much better explanation of the rationale for their selection. It is possibly related to what may be understandable difficulties with a challenging geochemical method, although others, notably Hall, also Negre, McManus, Lippold and Böhm have demonstrated that it is possible to produce continuous highly resolved records of the same isotope systems for specific intervals. Or it may be related to the

quality or continuity of the sediment cores. These are acceptable reasons if they are confronted and explained, although it would be most satisfactory if some greater level of scientific rationale were presented. This is currently inadequate, beyond the mention of an interval that was not included. A section of a paragraph or two that would better explain the reasons for the scattered data intervals might seem to the authors to be an acknowledgement of a shortcoming, but in the end it would increase the interest and potential impact of the published study.

The issue of chronology may be even more crucial, as the authors draw potentially important conclusions about intervals that do not appear to coincide with their data exactly, or in one crucial instance, at all. Figure 3 makes this very clear. None of the shaded intervals truly represent interstadials. The red shaded intervals all cover some portion of one interstadial or another, but the oldest begins at the peak of GI10 and extends beyond the peak of the next stadial, the subsequent shading covers solely a portion of the transition from GI 8 to the next stadial, without including the interstadial peak at all, and the youngest of the three is the only one to cover the entire interstadial GI3, but also includes two times as much duration of full stadial conditions. This does not appear to be just a drafting issue, which might be easily remedied. The shading is well aligned with the sediment data, which largely do not coincide with the ice core evidence. In the case of the fourth time slice, HS2, the blue shading in Figure 3 aligns well with the new data, until there is an abrupt data gap above the most extreme values, apparently due to a turbidite layer. But the shaded interval is centered on 26 ka, when the published age for HS2 is more than one to two thousand years younger (Naafs et al., 2013, Hodell et al., 2008, Hemming, 2004). Because this interval is well dated, it seems that the new data are older than HS2, which might instead correspond and even be related to the turbidite interval. A related question is how there appear to be data from this same interval, which is presented as a several thousand year gap in the supplemental figure S2.

The authors very reasonably identified intervals of stability in the circulation based

on their data, to make the most informative comparison with the model results. These choices did not lead to direct comparisons with the Greenland climate variations, which they accurately describe as important intervals for which the past circulation is not fully or well understood At the very least these chronological issues should be confronted. If they can be adjusted or adequately explained, it will greatly enhance the significance of this study.

Specific comments-

As mentioned in the introduction, the 13C data should have complications due to carbon cycling as well as ocean circulation. These can also be better addressed when interpreting the different time slices, and may help to explain differences in the data not due to circulation.

The authors describe an important change at the onset of HS2. Aside from the chronological issues, do they infer that the observed changes relate only to the HS2 interval, or do they establish the LGM condition that is the focus of so many studies?

If it was only during HS2, was the configuration and strength then different from LGM?

The changes at various depths appear to be under-constrained by the data, in particular because some time slices utilize four sites and others more, but never more than six locations, and no two time slices utilize the same set of locations. This limits the confidence bounds possible in the interpretations, and must allow other consistent alternatives, which should be mentioned and possibly discussed.

The contrast between the inferred interstadial mode and HS2 mode appears to be related to which direction the waters were moving below 2500 meters. Does that mean that the deep Atlantic was influenced by southern source waters below 2500 in both scenarios?

Many schematic and model representations of the deep Atlantic display a boundary between northern and southern waters that is inclined as a function of latitude. Do the

authors consider that also to be possible in their reconstructions?

The presented model shows that boundary to slope deeper to the south in the Holocene, which might suggest that northern waters influence more of the volume of the south Atlantic than the north. Perhaps this can be explained and clarified for those less familiar with this type of geochemical modeling.

Is the southward flowing mass at intermediate depth GNAIW? Several studies mentioned have inferred a vigourous circulation by this water mass, at least at the LGM. The contrasting conclusion of a sluggish intermediate circulation here is largely based on 13C from the productive equatorial region. Nevertheless, it would be useful to have a more direct discussion in the context of previous interpretations.

Technical points with page and line numbersp4 l4 THE GeoB3910 age model

p10 l25 values measured in the deep equatorial core during HS2 ARE consistent

Fig S4 Two panels compare similar data, yet are presented in very different ways.

Fig S4, S5 Should show r2 values.

---

## Referee Comment (RC2) · R. Francois (Referee) · 6 Jun 2016

R. Francois (Referee)

rfrancois@eos.ubc.ca

Burckel et al combine new and published sediment Pa/Th and benthic d13C data with 2D simulations to assess the strength and geometry of the AMOC during 3 GIs and HS2 They chose these time intervals because they represent time periods with different ice sheet volumes

Their main conclusions are that AMOC during GIs consisted of a shallow northern overturning cell (likely weaker than the modern NADW) in the upper 2500m, above a deeper southern overturning cell whose volume flow would have been higher than modern AABW. During HS2, as per fig. 3, the circulation geometry stayed the same but was significantly more sluggish.

**CPD**

To me, the take-home message of this study is that the Atlantic overturning circulation during glacial climatic extrema (i.e. Greenland intertadials and Heinrich stadials) had a similar geometry, and were differentiated only by the strength of the overturning cells, with stronger overturning cells during Greenland interstadials and weaker ones during Heinrich stadials. What may be the most surprising here is the apparent stability of AMOC geometry through the glacial period. However, I don't think that circulation contrast between Greenland Interstadials and Greenland (non-Heinrich) Stadials has been clearly documented and discussed in the present manuscript.

General comments

As indicated by the authors, the complete interpretation of sediment Pa/Th will require, to the extent possible, a synoptic database for each time slice of interest. The present study is a valuable contribution towards this end, but I have some questions and comments regarding some details of the interpretation of the data.

Although I recommend "major revisions", I don't think that the revisions I suggest are "major". However, as I am very interested in the topic, I would like to have the opportunity to see the replies of the authors.

Comparing sediment Pa/Th and the Greenland temperature record.

If we accept that abrupt temperature changes in Greenland result from variations in heat transport coinciding with changes in the strength/geometry of the AMOC, one would not expect that changes in sediment Pa/Th would be concurrent with Greenland temperature changes. This is because of the response time of sediment Pa/Th to changes in circulation. For any abrupt change in overturning, the concentration of Pa and Th in the water column will adjust with an e-folding time equivalent to their residence time in the water column (ca. 100-200 y for Pa). It would thus take > 500 y to fully express the change in circulation in sedimentary Pa/Th. This may, in part, address the second question of the other reviewer, at least for GI 8 and 10. I suspect that GI3 may be too brief to yield a measureable Pa/Th signal. On the other hand, if

d13C is truly a water mass tracer, then we would expect much less or no lag between the 13C signal and Greenland temperature. However, if decreases in d13C are due to accumulation of nutrients resulting from a sluggish circulation, we would also expect a lag. Another complication when comparing sediment circulation proxies with Greenland temperature is that the latter may also be modulated by the location of the site of deep water formation. Particularly striking is the lack of a Greenland temperature signal at the transition between GS3 and HS2 (as is the case between LGM and HS1). In fact, the present manuscript does not address another key question which is whether there are noticeable changes in AMOC between Greenland Stadials and Interstadials (they only contrast Greenland Interstadials and Heinrich Stadials). I would argue that Pa/Th distribution reported to GI3 is mostly a Greenland Stadial signal (because of the brevity of GI3), suggesting no or little changes in AMOC between Greenland Stadials and Interstadials. If this is the case, abrupt changes in Greenland temperature could reflect changes in the site of deep water formation, or northward transport of cooler/warmer surface water. This question could probably be directly addressed with another time slice to the discussion.

Additional comments

Abstract; Line 21: "At the onset of HS2, the structure of the AMOC significantly changes" "Structure" is too vague a term. I think it is worth highlighting here that the present data set is interpreted to indicate that the geometry of the overturning circulation did not change (as per Fig. 7) but circulation was much weaker.

P2; line 4-5: ".. suggesting that other mechanisms could be required to explain Greenland temperature millennial scale variability" The accepted mechanism is heat transport by the AMOC. The presence of a shallow circulation cell during HS is not inconsistent with this mechanism and does not require an alternative explanation.

P2; line 14 – 15: I would suggest: However, interpretation of sediment Pa/Th from a single core can be ambiguous because similar values can result from different geometry and overturning strength (Luo et al., 2010). Reconstructing past circulation thus requires combining Pa/Th records from multiple sites over a wide range of latitudes and depths (refs).

P2; line 22: I would suggest: The streamfunction under Heinrich Stadial conditions* were simulated with the Earth System model Iloveclim (ref) while the Holocene stream-function was derived from geostrophic velocity estimates (ref) *later on, this is becoming confusing, since the HS simulation does not fit the HS data..

P3; line 6 – 8: This should be moved to section 2.1.2. I note that the South Atlantic record of Jonkers et al is not mentioned. Is it because this core sedimentation rate is to low? It would still be worth checking if their glacial values are consistent with the AMOC scenarios presented here.

P3; line 34: "Pa/Th records renewal rates of water masses ca. 1000m above the seafloor" While it is correct that sediment Pa/Th records Pa and Th scavenging mostly coming from the water ca. 1000m above the seafloor, it does not record renewal rates of this water mass. The scavenging of Pa and Th from this water mass is in part controlled by its Pa and Th concentration, which is influenced by the overall geometry and strength of the AMOC. That is why, as indicated by the authors, interpretation of sediment Pa/Th requires a synoptic database for each time slice of interest.

P3; line 40 I suggest "High (low) rate of overturning" rather than "High (low) flow rates.." If circulation was only horizontal and scavenging intensity uniform in the ocean, sediment Pa/Th would not be dependent on flow rate

P5; line 13: I would suggest: ..Pa/Th increases along the flow path of any newly-formed deep water masses*, as initially low dissolved Pa concentrations increase... *it is not true that Pa/Th increase along the flow path of any water mass.

P5; line 18-19: While there is no explicit parameterization of diffusive transport in the 2D model, it is present in the model and it is controlled by horizontal velocities and

horizontal grid spacing. In the model used by Luo et al., the inherent mixing is about 800ms$-2$, which is in the upper range of the along-isopycnal tracer diffusivities. Therefore, it is not the lack of diffusive transport that prevents the model from simulating boundary scavenging. Instead, it is simply because it is a 2D model and there are no margins. Including boundary scavenging at ocean margins would require a 3D model (or an open 2D model).

P5; line 21 – 22: Boundary scavenging is weak in the Holocene Atlantic because of the short residence time of deep water in this basin (which results from a high overturning rate). This may not be the case for Heinrich Stadials and the expression of boundary scavenging at the margins during these events would depend on their duration. If the ocean stays in its Heinrich Stadial mode long enough (500 – 1000 years?) to start expressing boundary scavenging, the 2 D model will overestimate the Pa/Th in cores located in low productivity central basin regions and underestimate the Pa/Th in cores located at the margin. This needs to be kept in mind when interpreting the data

P5; line 35: As discussed above, if their duration is long enough, boundary scavenging should be expressed during Heinrich Stadials (if they are characterized by a very sluggish AMOC). If it is expressed during H4 but not during H2, this is an observation that needs discussion (was AMOC more sluggish during HS4? Was HS2 a briefer event? These questions should at least be raised). On the other hand, based on Fig. 3, it seems that boundary scavenging was also expressed during HS2 (as we would expect..)

Fig. 1: I suggest adding a panel showing long/lat of the cores to make it easier to visualize how boundary scavenging could affect Pa/Th in each core

Fig. 3 caption: I don't understand "average Pa/Th for each core is represented by the lines"

P6; line 21: I would remove "indicating the absence of Pa export" Instead, Pa/Th > 0.093 indicates the influence of boundary scavenging in this margin core. We would

then expect that the 2D model underestimate the measured Pa/Th. Likewise, we would expect that Pa/Th measured in open ocean cores during that time would be lower than those generated by the model.

P6; line 22: "Pa/Th variability associated with GS and GI is observed" Pa/Th for GI 10, 8 and HS2 (and 4; I am not sure why HS4 is not considered in the discussion; boundary scavenging is also apparent during HS2) are well documented. If the authors want to discuss AMOC variability between GS and GI, however, they need to add and discuss another time slice corresponding to a GS (same remark for p7; line 19)

Section 4.2.1 GI data fit well with the HS1 simulation (particularly is the latitude of deep water formation is adjusted). On the other hand, HS2 data do not fit well with HS1 simulation. This is confusing. If we accept the interpretation of the HS2 data, that would mean that the so-called HS1 simulation does not simulate circulation during Heinrich Stadials. Shouldn't then this simulation be called something else? (e.g. shallow, moderate overturning circulation scheme or such). What is the basis for taking the "HS1" streamfunction as representative of Heinrich Stadial circulation?

P11; line 33: "Our data shows that the geometry of the AMOC changed at the onset of HS2" As illustrated on Fig. 7, the geometry did not change, only the rate of volume transport changed.

S2 (Pa/Th uncertainties) I don't understand the meaning of "Hence, Pa/Th values associated with each time slice on core MD.. is invariant, despite dating uncertainties"

---

## Author Comment (AC1) · 3 Jul 2016

Full point by point response to reviewers' comments on manuscript "Changes in the geometry and strength of the Atlantic Meridional Overturning Circulation during the last glacial (20-50 ka)".

We would like to thank the reviewers for their constructive comments. Our point by point response is outlined below. The reviewer's comments are displayed, and our answers are highlighted by asterisks "***". As requested by the editor, we will provide a revised version of the manuscript later in the revision process. Note that page and line numbers that we provide are those associated with the PDF downloaded from http://www.clim-past-discuss.net/cp-2016-26/#discussion. The line numbers the first reviewer provided

in the "Technical points" section appear to be different from those that appear on the PDF. Thank you for your understanding.

Referee #1 (Anonymous)

In their manuscript "Changes in the geometry and strength of the Atlantic Meridional Overturning Circulation during the last glacial (20-50 ka)", Burckel et al. use 231Pa and 230Th ratios and 13C to assess the past state of deep ocean circulation in the Atlantic Ocean at several intervals during the past glaciation. After attempting to assess the geometry and strength of the overturning cell of the Atlantic, they conclude that the deep ocean circulation was very different from the modern in all four of their study intervals. The interstadial circulation was different in being relatively shallow, with a deep inflow from the south. Southward flowing waters at mid-depth would therefore have been the return flow of southern-sourced waters. At the time of Heinrich Stadial 2, yet another different circulation is inferred, with southern waters filling the deep Atlantic and a slow, southward-flowing water mass occupying the intermediate depths. This is a potentially valuable contribution to the literature on past states of the ocean circulation. It presents new geochemical data in a spatial array that may provide insights into changes at different depths and locations. The isotopic method is a promising and exciting approach, although it seems still in development in comparison to modern measurements. The data are compared to model output, which although limited in resolution and lacking a third dimension, nevertheless provides useful constraints on potential interpretations. The conclusions are not inconsistent with the relatively limited data presented. In terms of the specific criteria, the paper certainly addresses relevant questions within the scope of CP. It does not present novel approaches, but builds well upon existing techniques, data, and ocean modeling output. Substantial conclusions are reached regarding the configuration and rate of ocean circulation. The conclusions are not inconsistent with the data, although there are too many gaps at relevant locations and depths for them to be any more convincing than many alternatives which are

not discussed. Figures are relatively clear, and text is a reasonable length. The text is fluent and the authors give adequate credit to the previous studies that they utilize and discuss. The two largest issues with the paper in its present form are related to its justification and chronology. This is a study of four time slices that are widely distributed within the last glacial. They are neither the most extreme, nor the most characteristic. Nor do they include important transitions or intervals of special climatic interest. It is therefore not clear to the reader why this seemingly arbitrary assortment of time slices was chosen. The authors should provide a much better explanation of the rationale for their selection. It is possibly related to what may be understandable difficulties with a challenging geochemical method, although others, notably Hall, also Negre, McManus, Lippold and Böhm have demonstrated that it is possible to produce continuous highly resolved records of the same isotope systems for specific intervals. Or it may be related to the quality or continuity of the sediment cores. These are acceptable reasons if they are confronted and explained, although it would be most satisfactory if some greater level of scientific rationale were presented. This is currently inadequate, beyond the mention of an interval that was not included. A section of a paragraph or two that would better explain the reasons for the scattered data intervals might seem to the authors to be an acknowledgement of a shortcoming, but in the end it would increase the interest and potential impact of the published study.

\*\*\* Pa/Th measurements were focused on relevant MIS3 time slices. HS2 and HS4 in particular were selected because these intervals are characterized by significantly different ice sheet volumes (Lambeck and Chappell, 2001) (see P.2, l.23 of the manuscript). Oceanic circulation around these time periods could therefore reasonably be expected to be different. Unfortunately, it is difficult to disentangle the sedimentary from the oceanic influences on the Pa/Th signal during HS4 in core MD09-3257, as high Pa/Th values are correlated to high 232Th fluxes (Burckel et al., 2015). We therefore focused our study on the time intervals during which the Pa/Th signal of core MD09-3257 can be interpreted in terms of circulation changes, i.e. HS2 and on the DO climate variability encompassing HS2 and HS4. \*\*\*

The issue of chronology may be even more crucial, as the authors draw potentially important conclusions about intervals that do not appear to coincide with their data exactly, or in one crucial instance, at all. Figure 3 makes this very clear. None of the shaded intervals truly represent interstadials. The red shaded intervals all cover some portion of one interstadial or another, but the oldest begins at the peak of GI10 and extends beyond the peak of the next stadial, the subsequent shading covers solely a portion of the transition from GI 8 to the next stadial, without including the interstadial peak at all, and the youngest of the three is the only one to cover the entire interstadial GI3, but also includes two times as much duration of full stadial conditions. This does not appear to be just a drafting issue, which might be easily remedied. The shading is well aligned with the sediment data, which largely do not coincide with the ice core evidence.

\*\*\* In figure 3, it is clear that every GI (in particular GI10, 8 and 7) is associated with a Pa/Th decrease (i.e. increased circulation intensity). The GI8 and GI10 time slices are well defined as periods of stable oceanic circulation (see section 2.3). Because GI3 is of shorter duration, it is possible that the GI3 time slice does not represent average interstadial conditions, as highlighted page 6, line 10 (see also comments from and answers to Roger François, 2nd reviewer). \*\*\*

In the case of the fourth time slice, HS2, the blue shading in Figure 3 aligns well with the new data, until there is an abrupt data gap above the most extreme values, apparently due to a turbidite layer. But the shaded interval is centered on 26 ka, when the published age for HS2 is more than one to two thousand years younger (Naafs et al., 2013, Hodell et al., 2008, Hemming, 2004). Because this interval is well dated, it seems that the new data are older than HS2, which might instead correspond and even be related to the turbidite interval.

\*\*\* It is important to distinguish the Heinrich Stadial (defined as the stadial (cold) period during which a Heinrich Event (HE) occurs) and the event itself, characterized by the sedimentary IRD layers. The Pa/Th increase that we observe and that is concurrent

with the increase observed in core ODP1063 occurs during GS2, the cold period in the Greenland temperature record during which HE2 is observed in marine sediment cores. \*\*\*

A related question is how there appear to be data from this same interval, which is presented as a several thousand year gap in the supplemental figure S2.

\*\*\* Turbiditic layers were identified in core MD09-3256Q between 24.16 and 20.88 ka (gap in Figure S2). However, no sedimentary Pa/Th data from this core corresponding to this interval are presented in Figure 3 (last Pa/Th data at 24.16 ka). \*\*\*

The authors very reasonably identified intervals of stability in the circulation based on their data, to make the most informative comparison with the model results. These choices did not lead to direct comparisons with the Greenland climate variations, which they accurately describe as important intervals for which the past circulation is not fully or well understood At the very least these chronological issues should be confronted. If they can be adjusted or adequately explained, it will greatly enhance the significance of this study.

\*\*\* The fact that oceanic and Greenland signals do not align perfectly could be due to (i)-chronological uncertainties (ii)-real leads or lags of one signal compared to the other (iii)-the response time of geochemical proxies to changes in oceanic circulation. Note that chronological uncertainties were accounted for in calculating the uncertainties associated with the Pa/Th values of each time slice (see supplementary material). \*\*\*

Specific comments-

As mentioned in the introduction, the 13C data should have complications due to carbon cycling as well as ocean circulation. These can also be better addressed when interpreting the different time slices, and may help to explain differences in the data not due to circulation.

*** We lack data on changes in marine productivity at the studied sites so we cannot investigate what fraction of the benthic d13C might reflect these changes. We thus follow the classical assumption that d13C reflects changes in bottom water ventilation. ***

The authors describe an important change at the onset of HS2. Aside from the chronological issues, do they infer that the observed changes relate only to the HS2 interval, or do they establish the LGM condition that is the focus of so many studies? If it was only during HS2, was the configuration and strength then different from LGM?

*** The change that we observe at the onset of HS2 in core MD09-3257 specifically relates to the HS2 interval, as we observe an increased Pa/Th at the beginning of GS2. Based on Pa/Th and d13C data in cores MD09-3257 and GeoB3910, the onset of the LGM appears to be characterized by an active circulation, however not as active as that of the Holocene. ***

The changes at various depths appear to be under-constrained by the data, in particular because some time slices utilize four sites and others more, but never more than six locations, and no two time slices utilize the same set of locations. This limits the confidence bounds possible in the interpretations, and must allow other consistent alternatives, which should be mentioned and possibly discussed.

*** We agree with reviewer #1's comment and added the following two sentences at the end of section 2.4 to clarify our argumentation: "Note that due to the limited number of sedimentary Pa/Th records during MIS3, we can only provide an approximate estimate of water mass boundary positions. Our equatorial transect is however ideally located to record shifts in the position of the transition between southern and northern sourced water masses." ***

The contrast between the inferred interstadial mode and HS2 mode appears to be related to which direction the waters were moving below 2500 meters. Does that mean that the deep Atlantic was influenced by southern source waters below 2500 in both

scenarios?

\*\*\* Based on our results, we infer that the Atlantic was likely influenced by southern sourced waters below 2500 m during HS2 but we lack data to determine the precise vertical extent of this southern-sourced water mass. In contrast, during Greenland Interstadials, the transition between southern- and northern-sourced water masses was probably located between 3500 and 2500 m, which would explain the low Pa/Th gradient between our equatorial sediment cores. \*\*\*

Many schematic and model representations of the deep Atlantic display a boundary between northern and southern waters that is inclined as a function of latitude. Do the authors consider that also to be possible in their reconstructions?

\*\*\* The models representing the deep Atlantic (i.e. streamfunctions, Fig.2, b, d, f), do not display an inclined boundary as a function of latitude. However, the simulated sedimentary Pa/Th (Fig.2, a, c, e) do show increasing sedimentary Pa/Th with latitude along the flow path of any newly formed water mass. We explain this effect page 5 line 11. \*\*\*

The presented model shows that boundary to slope deeper to the south in the Holocene, which might suggest that northern waters influence more of the volume of the south Atlantic than the north. Perhaps this can be explained and clarified for those less familiar with this type of geochemical modeling.

\*\*\* We are afraid we do not fully understand this question. To render our argumentation accessible to the non-specialized audience, we describe the behavior of dissolved Pa and Th and how this influences the output of the model (see section 2.1.2 and 2.2.1). For a more thorough explanation, we refer the reader to the chapter book by Francois, 2007 (main principles of Pa/Th as a proxy of oceanic circulation intensity) and to the Luo et al., 2010 paper (description of the 2D Pa/Th model). \*\*\*

Is the southward flowing mass at intermediate depth GNAIW? Several studies mentioned have inferred a vigourous circulation by this water mass, at least at the LGM. The contrasting conclusion of a sluggish intermediate circulation here is largely based on 13C from the productive equatorial region. Nevertheless, it would be useful to have a more direct discussion in the context of previous interpretations.

*** We make sure not to describe the southward flowing water as GNAIW, as it is indeed defined for the LGM and our study concerns earlier time periods. Our conclusion concerning the sluggish intermediate water mass only relates to HS2. We then see a decrease in Pa/Th (i.e. likely an increase in the overturning intensity) at the onset of the LGM.

Also, we made a few minor changes in order to make clear that we were careful not to over-interpret benthic d13C:

P.5, l.28: the sentence "Moreover, benthic foraminiferal d13C measurements, which reflect the DIC of the water mass directly above the sediment interface, allows confirming or infirming the geometry information contained in measured Pa/Th values." was removed.

P.8, l.30: the sentence "the high d13C values of core SU90-03 and MD09-3257 indicate that northern sourced waters were present at ∼2500 m in the North and equatorial Atlantic" was changed to "the high d13C values of core SU90-03 and MD09-3257, and low d13C values of core MD02-2594 (< 0.5‰ Negre et al., 2010), indicate that northern sourced waters were present at ∼2500 m in the North and equatorial Atlantic".

P.11, l.13: "deep waters likely dominated the deep Atlantic Ocean" was replaced by "deep waters likely filled the deep Atlantic Ocean". We also removed the word "direct" in the sentence : "The direct influence of the southern-sourced water mass likely extended...".

P.11, l.17: we removed "and their associated return flow" from the sentence "...it is difficult to assess the exact position of the southern sourced waters and their associated

return flow."

P.11, l.17: we removed the word "directly" in the sentence "This water mass probably directly affected the equatorial Atlantic. . .".***

---

## Author Comment (AC2) · 3 Jul 2016

Full point by point response to reviewers' comments on manuscript "Changes in the geometry and strength of the Atlantic Meridional Overturning Circulation during the last glacial (20-50 ka)".

We would like to thank the reviewers for their constructive comments. Our point by point response is outlined below. The reviewer's comments are displayed, and our answers are highlighted by asterisks "***". As requested by the editor, we will provide a revised version of the manuscript later in the revision process. Note that page and line numbers that we provide are those associated with the PDF downloaded from http://www.clim-past-discuss.net/cp-2016-26/#discussion. The line numbers the first reviewer provided

in the "Technical points" section appear to be different from those that appear on the PDF. Thank you for your understanding.

Referee #2 (R. Francois)

Burckel et al combine new and published sediment Pa/Th and benthic d13C data with 2D simulations to assess the strength and geometry of the AMOC during 3 GIs and HS2 They chose these time intervals because they represent time periods with different ice sheet volumes. Their main conclusions are that AMOC during GIs consisted of a shallow northern overturning cell (likely weaker than the modern NADW) in the upper 2500m, above a deeper southern overturning cell whose volume flow would have been higher than modern AABW. During HS2, as per fig. 3, the circulation geometry stayed the same but was significantly more sluggish. To me, the take-home message of this study is that the Atlantic overturning circulation during glacial climatic extrema (i.e. Greenland intertadials and Heinrich stadials) had a similar geometry, and were differentiated only by the strength of the overturning cells, with stronger overturning cells during Greenland interstadials and weaker ones during Heinrich stadials. What may be the most surprising here is the apparent stability of AMOC geometry through the glacial period. However, I don't think that circulation contrast between Greenland Interstadials and Greenland (non-Heinrich) Stadials has been clearly documented and discussed in the present manuscript.

General comments As indicated by the authors, the complete interpretation of sediment Pa/Th will require, to the extent possible, a synoptic database for each time slice of interest. The present study is a valuable contribution towards this end, but I have some questions and comments regarding some details of the interpretation of the data. Although I recommend "major revisions", I don't think that the revisions I suggest are "major". However, as I am very interested in the topic, I would like to have the opportunity to see the replies of the authors.

[Figure]

Comparing sediment Pa/Th and the Greenland temperature record. If we accept that abrupt temperature changes in Greenland result from variations in heat transport coinciding with changes in the strength/geometry of the AMOC, one would not expect that changes in sediment Pa/Th would be concurrent with Greenland temperature changes. This is because of the response time of sediment Pa/Th to changes in circulation. For any abrupt change in overturning, the concentration of Pa and Th in the water column will adjust with an e-folding time equivalent to their residence time in the water column (ca. 100-200 y for Pa). It would thus take > 500 y to fully express the change in circulation in sedimentary Pa/Th. This may, in part, address the second question of the other reviewer, at least for GI 8 and 10. I suspect that GI3 may be too brief to yield a measureable Pa/Th signal. On the other hand, if d13C is truly a water mass tracer, then we would expect much less or no lag between the 13C signal and Greenland temperature. However, if decreases in d13C are due to accumulation of nutrients resulting from a sluggish circulation, we would also expect a lag. Another complication when comparing sediment circulation proxies with Greenland temperature is that the latter may also be modulated by the location of the site of deep water formation. Particularly striking is the lack of a Greenland temperature signal at the transition between GS3 and HS2 (as is the case between LGM and HS1).

*** We do not consider d13C as a perfect water mass tracer. The d13C of benthic foraminifera C. wuellerstorfi is a proxy of the nutrient content of bottom water masses, that we interpret as reflecting bottom water ventilation. For instance, reduced d13C at a site influenced by northern sourced waters could result from increased southern sourced water mass influence, or reduced deep water formation in the North Atlantic region. However, we would like to stress that we are not trying to resolve the timing between changes in deep water circulation and Greenland climate. Timing issues do not alter the interpretation of our time slices, as they are defined based on stable oceanic conditions during Greenland interstadials. ***

In fact, the present manuscript does not address another key question which is whether

there are noticeable changes in AMOC between Greenland Stadials and Interstadials (they only contrast Greenland Interstadials and Heinrich Stadials). I would argue that Pa/Th distribution reported to GI3 is mostly a Greenland Stadial signal (because of the brevity of GI3), suggesting no or little changes in AMOC between Greenland Stadials and Interstadials. If this is the case, abrupt changes in Greenland temperature could reflect changes in the site of deep water formation, or northward transport of cooler/warmer surface water. This question could probably be directly addressed with another time slice to the discussion.

*** Because they span different depths on the Brazilian margin, cores MD09-3257 and MD09-3256Q Pa/Th records are particularly interesting to understand the geometry and strength of the AMOC during MIS3. Unfortunately we lack data in core MD09-3256Q during Greenland stadials. We therefore decided not to define stadial time slices. However, as we point out page 6, line 10, we agree that GI3 time slice might not reflect interstadial conditions. We have therefore added the following short paragraph (inserted P.9, l.20) to explain that this time slice may reflect stadial conditions and discuss the implications: "Based on our definition of Interstadial time slices, we assume that the GI3 time slice reflects interstadial conditions. However, because GI3 seen in Greenland ice cores is of relatively short duration, the Pa/Th signal of the studied sediment cores might not reflect full interstadial circulation conditions. Nonetheless, we consider it unlikely that the Pa/Th of GI3 reflects stadial conditions. Indeed, core MD09-3257 sedimentary Pa/Th values observed during GI3 are similar to those recorded during the GI8 and GI10 time slices that correspond to strict interstadials (Fig. 3).". ***

Additional comments Abstract; Line 21: "At the onset of HS2, the structure of the AMOC significantly changes" "Structure" is too vague a term. I think it is worth highlighting here that the present data set is interpreted to indicate that the geometry of the overturning circulation did not change (as per Fig. 7) but circulation was much weaker.

*** While this is indeed a possibility, we do not conclude that the geometry of the AMOC did not change between Heinrich Stadial 2 and Greenland Interstadials. We cannot

say if the southern sourced water mass influence extended above 2500 m during HS2 (page 11, line 13). We removed the term structure and wrote "At the onset of Heinrich Stadial 2, the AMOC intensity and geometry likely changed". \*\*\*

P2; line 4-5: ".. suggesting that other mechanisms could be required to explain Greenland temperature millennial scale variability" The accepted mechanism is heat transport by the AMOC. The presence of a shallow circulation cell during HS is not inconsistent with this mechanism and does not require an alternative explanation.

\*\*\* We agree with your point and therefore chose to use the terms "could be required". In order to clarify the text, we modified the sentence as follows "... suggesting that Greenland temperature millennial scale variability might be related to more complex changes in Atlantic circulation than simply switching between "on" and "off" circulation modes." \*\*\*

P2; line 14 – 15: I would suggest: However, interpretation of sediment Pa/Th from a single core can be ambiguous because similar values can result from different geometry and overturning strength (Luo et al., 2010). Reconstructing past circulation thus requires combining Pa/Th records from multiple sites over a wide range of latitudes and depths (refs).

\*\*\* Your suggestion makes the issue of interpreting a single core clearer by pointing the possibility of having multiple circulation intensities for a single sedimentary Pa/Th value. We modified the sentence following your suggestion. \*\*\*

P2; line 22: I would suggest: The streamfunction under Heinrich Stadial conditions\* were simulated with the Earth System model Iloveclim (ref) while the Holocene streamfunction was derived from geostrophic velocity estimates (ref) \*later on, this is becoming confusing, since the HS simulation does not fit the HS data..

\*\*\* We changed the sentence to: "One streamfunction is derived from present day geostrophic velocity estimates (Talley et al., 2003) and two others were simulated with

the Earth System model iLOVECLIM under different climatic conditions (Roche et al., 2014)." As described below, we agree with your comment on the confusing nature of the terminology used and changed "HS1 streamfunction" into "Shallow overturning streamfunction". ***

P3; line 6 – 8: This should be moved to section 2.1.2.

*** Although we understand the potential issue of mentioning Pa/Th in the "Sediment cores" section, we prefer to group all cores at the beginning of the paper for the sake of clarity. ***

I note that the South Atlantic record of Jonkers et al is not mentioned. Is it because this core sedimentation rate is to low? It would still be worth checking if their glacial values are consistent with the AMOC scenarios presented here.

*** Unfortunately, none of Jonkers et al.'s Pa/Th data are within our time slices. However, the low sedimentary Pa/Th values that they describe fit very well with our assumptions of intensified deep water formation in the South Atlantic (it would help to systematically exclude the Holocene streamfunction). ***

P3; line 34: "Pa/Th records renewal rates of water masses ca. 1000m above the seafloor" While it is correct that sediment Pa/Th records Pa and Th scavenging mostly coming from the water ca. 1000m above the seafloor, it does not record renewal rates of this water mass. The scavenging of Pa and Th from this water mass is in part controlled by its Pa and Th concentration, which is influenced by the overall geometry and strength of the AMOC. That is why, as indicated by the authors, interpretation of sediment Pa/Th requires a synoptic database for each time slice of interest.

*** We agree that the sentence: "Pa/Th is a relatively recent tracer that records the renewal rate of water masses occupying the first ∼1000 m above the seafloor (Thomas et al., 2006)" could be misleading and we replaced it by "Pa/Th is a relatively recent tracer that can be used to estimate the renewal rate of water masses occupying the

first ∼1000 m above the seafloor (Thomas et al., 2006, Luo et al., 2010)". ***

P3; line 40 I suggest "High (low) rate of overturning" rather than "High (low) flow rates.."If circulation was only horizontal and scavenging intensity uniform in the ocean, sediment Pa/Th would not be dependent on flow rate

*** We agree that overturning is indeed required for sedimentary Pa/Th. We changed the sentence following your suggestion. ***

P5; line 13: I would suggest: ..Pa/Th increases along the flow path of any newly-formed deep water masses*, as initially low dissolved Pa concentrations increase: : : *it is not true that Pa/Th increase along the flow path of any water mass.

*** We agree and changed the sentence following your suggestion. ***

P5; line 18-19: While there is no explicit parameterization of diffusive transport in the 2D model, it is present in the model and it is controlled by horizontal velocities and horizontal grid spacing. In the model used by Luo et al., the inherent mixing is about 800msôĂĂ2, which is in the upper range of the along-isopycnal tracer diffusivities. Therefore, it is not the lack of diffusive transport that prevents the model from simulating boundary scavenging. Instead, it is simply because it is a 2D model and there are no margins. Including boundary scavenging at ocean margins would require a 3D model (or an open 2D model).

*** Thank you very much for your input concerning the 2D Pa/Th model. We changed the paragraph into the following: : "The absence of margins in the simple 2D Pa/Th model (Luo et al., 2010) prevents it from simulating boundary scavenging, which is the transfer of dissolved protactinium from open ocean regions of high Pa concentrations to coastal regions of low Pa concentration such as in upwelling zones (Christl et al., 2010). However, as described in the results section, we verified that our Pa/Th signal is mainly driven by oceanic circulation changes and the importance of diffusive transport is therefore likely negligible here. This simple 2D Pa/Th model therefore appears

adequate for comparison with our Pa/Th data." ***

P5; line 21 – 22: Boundary scavenging is weak in the Holocene Atlantic because of the short residence time of deep water in this basin (which results from a high overturning rate). This may not be the case for Heinrich Stadials and the expression of boundary scavenging at the margins during these events would depend on their duration. If the ocean stays in its Heinrich Stadial mode long enough (500 – 1000 years?) to start expressing boundary scavenging, the 2 D model will overestimate the Pa/Th in cores located in low productivity central basin regions and underestimate the Pa/Th in cores located at the margin. This needs to be kept in mind when interpreting the data

*** See below for answers concerning these concerns. ***

P5; line 35: As discussed above, if their duration is long enough, boundary scavenging should be expressed during Heinrich Stadials (if they are characterized by a very sluggish AMOC). If it is expressed during H4 but not during H2, this is an observation that needs discussion (was AMOC more sluggish during HS4? Was HS2 a briefer event? These questions should at least be raised). On the other hand, based on Fig. 3, it seems that boundary scavenging was also expressed during HS2 (as we would expect..)

*** Yes, we indeed have boundary scavenging during HS2, e.g. indicated by sedimentary Pa/Th ratios above the Pa/Th production ratio. However, we showed in a previous study that changes in sedimentary Pa/Th are mainly driven by oceanic circulation changes in core MD09-3257 (Burckel et al., 2015). On the contrary, HS4 Pa/Th values in core MD09-3257 appear to be mostly driven by vertical terrigenous fluxes (Burckel et al., 2015). We therefore chose to exclude these values (open squares in Fig. 3) and HS4 when discussing oceanic circulation. ***

Fig. 1: I suggest adding a panel showing long/lat of the cores to make it easier to visualize how boundary scavenging could affect Pa/Th in each core

\*\*\* We agree with your comment, but table S1 already lists the cores' positions and depths, along with the age models used. We therefore prefer to add a second panel to Figure 1 to show the position of all cores in the Atlantic Ocean. \*\*\*

Fig. 3 caption: I don't understand "average Pa/Th for each core is represented by the lines"

\*\*\* We mean "Replicates averaged Pa/Th signal". We modified the sentence: "In (a) the average Pa/Th for each core is represented by the lines and individual measurements by diamonds or squares (MD09-3257)." into "In (a) lines pass through average Pa/Th values in case of replicates, while diamonds and squares (MD09-3257) correspond to individual Pa/Th measurements". \*\*\*

P6; line 21: I would remove "indicating the absence of Pa export" Instead, Pa/Th > 0.093 indicates the influence of boundary scavenging in this margin core. We wouldthen expect that the 2D model underestimate the measured Pa/Th. Likewise, we would expect that Pa/Th measured in open ocean cores during that time would be lower than those generated by the model.

\*\*\* We would like to keep this sentence, as high Pa/Th signal at that time is reflecting reduced overturning rates. If we were to write "indicating boundary scavenging", we fear that the reader might think that sedimentary processes are overprinting the oceanic circulation information. Moreover, because reduced Atlantic basin width at the latitude of our Brazilian sites could result in an overestimation of simulated sedimentary Pa/Th (Lippold et al., 2011, see p.9, l.33 of the manuscript), the underestimation of sedimentary Pa/Th due to the absence of boundary scavenging in the 2D model might be partially or totally compensated on the North Brazilian margin. \*\*\*

P6; line 22: "Pa/Th variability associated with GS and GI is observed" Pa/Th for GI 10, 8 and HS2 (and 4; I am not sure why HS4 is not considered in the discussion; boundary scavenging is also apparent during HS2) are well documented. If the authors want to discuss AMOC variability between GS and GI, however, they need to add and discuss

another time slice corresponding to a GS (same remark for p7; line 19)

\*\*\* Although there seems to be a difference between GI and GS oceanic circulation based on core MD09-3257 Pa/Th record we do not want to discuss variability between GS and GI because we lack a more comprehensive picture of the circulation during GS due to the absence of Pa/Th data in core MD09-3256Q during these periods. \*\*\*

Section 4.2.1 GI data fit well with the HS1 simulation (particularly is the latitude of deep water formation is adjusted). On the other hand, HS2 data do not fit well with HS1 simulation. This is confusing. If we accept the interpretation of the HS2 data, that would mean that the so-called HS1 simulation does not simulate circulation during Heinrich Stadials. Shouldn't then this simulation be called something else? (e.g. shallow, moderate overturning circulation scheme or such). What is the basis for taking the "HS1" streamfunction as representative of Heinrich Stadial circulation?

\*\*\* The names of the streamfunctions originate from the paper of Roche et al., 2014. The HS1 streamfunction is generated with a 0.16 Sv freshwater forcing in the Labrador Sea and allows for the presence of a shallow overturning cell, while a 0.35 Sv forcing results in the absence of deep-water formation in the high latitude North Atlantic (off-mode). However, we agree that calling one of the streamfunctions HS1 is confusing, especially when compared to the HS2 time slice. We therefore renamed this streamfunction "Shallow overturning streamfunction" in the entire manuscript. \*\*\*

P11; line 33: "Our data shows that the geometry of the AMOC changed at the onset of HS2" As illustrated on Fig. 7, the geometry did not change, only the rate of volume transport changed.

\*\*\* You are right, we modified "changed" into "likely changed". Based on our data we cannot determine the exact vertical extent of the southern sourced water mass on the Brazilian margin. We also modified Fig. 7 in order to picture the uncertainty on the vertical extent of the southern sourced water mass in the HS2 time slice. \*\*\*

S2 (Pa/Th uncertainties) I don't understand the meaning of "Hence, Pa/Th values associated with each time slice on core MD.. is invariant, despite dating uncertainties"

\*\*\* Because time slices are defined based on MD09-3257 Pa/Th signal, age uncertainties do not affect Pa/Th uncertainties associated with each time slice in this core. For all other cores, age uncertainties affect the Pa/Th uncertainties associated with each time slice. This has been clarified in the text. The new sentence reads: "Hence, Pa/Th values associated with the different time slices in core MD09-3257 are independent from the age model. For all other cores, dating uncertainties account for Pa/Th uncertainties associated with each time slice." \*\*\*

―――――――――――――――――――

[Figure]

[Figure]

**Fig. 1.** Figure 1

**(a)**

**Greenland Interstadials**

Southern sourced water, low $\delta^{13}$C

Northern sourced water, high $\delta^{13}$C

?

MD02-2594

MD09-3257

**5-20 Sv**

SU90-03

2300 m

**5-20 Sv**

MD09-3256Q

3500 m

ODP1063

4500 m

**(b)**

**Heinrich Stadial 2**

Southern sourced water, low $\delta^{13}$C

Northern sourced water, high $\delta^{13}$C

?

**weak overturning**

MD02-2594

MD09-3257

SU90-03

?

2300 m

MD09-3256Q

3500 m

ODP1063

4500 m

**Fig. 2.** Figure 7